# DendroTweaks, an interactive approach for unraveling dendritic dynamics

**Roman Makarov[1,2], Spyridon Chavlis[1], Panayiota Poirazi[1]***

[1]Institute of Molecular Biology and Biotechnology (IMBB), Foundation for Research and Technology-Hellas (FORTH), Heraklion, Greece; [2]Department of Biology, University of Crete, Heraklion, Greece

## eLife Assessment

Computational simulation of neuron function depends on a collection of morphological properties and ion channel biophysics. This manuscript introduces DendroTweaks, a **valuable** web application and Python library that eases interactive exploration, development, and validation of single-neuron models in an easily installable and well-documented package. The authors provide a **convincing** demonstration that their software aids with building intuition and rapid prototyping of biophysical models of neurons, which improves the accessibility of dendritic simulation.

**Abstract** Neurons rely on the interplay between two critical components, dendritic morphology and ion channels, to transform synaptic inputs into a sequence of somatic spikes. Detailed biophysical models with active dendrites have been instrumental in exploring this interaction. However, such models can be challenging to understand and validate due to the large number of parameters involved. In this work, we introduce *DendroTweaks*, a toolbox designed to make detailed biophysical models with active dendrites more intuitive and more interactive. *DendroTweaks* features a web-based graphical interface, where users can explore single-cell neuronal models and adjust their morphological and biophysical parameters with real-time visual feedback. In particular, *DendroTweaks* focuses on subcellular properties, such as kinetics and distribution of ion channels, as well as the dynamics and placement of synaptic inputs. The toolbox supports various experimental protocols designed to illuminate how morpho-electric properties map to dendritic events and how these dendritic events shape neuronal output, thereby enhancing model validation. It helps users build high-level, modular model representations and includes a rich set of tools for parsing, generating, and standardizing commonly used neuronal data formats. Finally, it enables model simplification through a built-in morphology reduction algorithm, allowing users to export models for further use in faster, more interpretable networks. By combining extensive visualization capabilities and comprehensive data management functionality, *DendroTweaks* introduces a novel interactive approach for unraveling dendritic dynamics. This approach will accelerate research on dendritic computations, their underlying mechanisms, and their fundamental role in brain function.

**\*For correspondence:**
poirazi@imbb.forth.gr

## Introduction

Neurons are the most well-studied brain cells, known for their key role in processing and storing information. Information travels from neuron to neuron via synaptic connections, which are typically formed on dendrites. These extensive branching processes of a neuron actively shape and transform synaptic inputs on their way to the soma, endowing neurons with a wide range of input-output transformations. Since Rall's pioneering work on signal propagation within dendrites (*Rall, 1959*), our understanding of dendritic dynamics has expanded with the discovery of local regenerative events,

such as Na$^+$ dendritic spikes (*Spencer and Kandel, 1961*; *Golding and Spruston, 1998*), and NMDA and Ca$^{2+}$ plateau potentials (*Schiller and Schiller, 2001*; *Llinas and Nicholson, 1971*; *Llinás and Hess, 1976*). These active properties of dendrites largely depend on the interaction between their branching morphology and ion channel composition. Multicompartmental biophysical models with active dendrites have been instrumental in exploring this relationship. However, the large number of parameters in such models complicates their interpretability, making them challenging to build, examine, and validate. In addition, the lack of standardization, along with their high computational complexity, makes these models less attractive for use in large-scale network simulations. As a result, state-of-the-art network models still consider dendrites as passive cables (*Markram et al., 2015*; *Billeh et al., 2020*), greatly underestimating their computational power (*Tran-Van-Minh et al., 2015*). At the same time, with the advent of new techniques like genetic tracking of ion channels and high-resolution imaging of neuronal activity using voltage-sensitive dyes, multicompartmental biophysical modeling is experiencing its renaissance. Detailed biophysical models, although mathematically complex and computationally inefficient, are ideal for capturing and explaining data produced, for example, through simultaneous voltage and synaptic input imaging in vivo.

These challenges and demands highlight the growing need to make biophysical neuronal models more accessible. Model accessibility can be considered on two complementary levels. First, there is *conceptual* accessibility, which refers to how well we can understand the system being modeled (e.g. *If I change parameter A, will outcome X follow*?). Second, there is *implementational* accessibility, which concerns the model as a computational artifact (e.g. *Can I easily change parameter A and measure X?*). Accessibility at the conceptual level can be improved through interactive hypothesis testing. This approach implies converting complex models into interactive visualizations that provide real-time feedback on how changes in morpho-electric parameters affect neuronal behavior. In particular, such functionality would help clarify how the ion channel kinetics and distribution shape dendritic events and somatic output. To our understanding, this is one of the biggest gaps in current neuronal modeling software. Interactive visualizations would also enhance model validation, shifting the focus from somatic spiking to activity throughout the cell. Finally, this approach would help identify which dendritic properties are essential for neuronal function and which can be discarded, enabling the simplification of single-cell models and their integration into faster, more interpretable networks. Accessibility at the implementational level can be improved by developing high-level, simulator-agnostic model representations that are capable of capturing complex dendritic properties. Such representations should be modular, enabling users to easily switch between stimulation protocols, morphologies, or parameter sets. In addition, accessibility at this level involves adopting best practices for data management and standardization. Together, these approaches result in more reusable, easier-to-understand models that are customizable according to users' needs.

In this work, we introduce *DendroTweaks*, a toolbox designed to make detailed biophysical models with active dendrites more accessible at both conceptual and implementational levels. Building on existing methods, we have developed a comprehensive workflow for developing single-cell models, including tuning their morphological and biophysical parameters, running stimulations, and analyzing the results. *DendroTweaks* is implemented as a Python package with an intuitive web-based graphical user interface (GUI). The GUI allows users to visually explore and fine-tune any parameter of the model, providing real-time feedback through interactive plots. The toolbox helps users build high-level, modular model representations that effectively capture the complex properties of active dendrites. Additionally, it includes a rich set of tools for parsing, generating, and standardizing commonly used neuronal data formats, facilitating interoperability with other neuronal modeling software. With *DendroTweaks*, users can better understand and control their models while exploring how dendritic properties shape neuronal activity. By making complex models more interpretable and interactive, *DendroTweaks* will advance research on the role of dendrites in brain function and deepen our understanding of these remarkable structures.

## Results

### Implementation and user interface

*DendroTweaks* is a Python toolbox designed for developing single-cell detailed biophysical models with active dendrites. It is inspired by the exploratory data analysis approach and allows any user, from

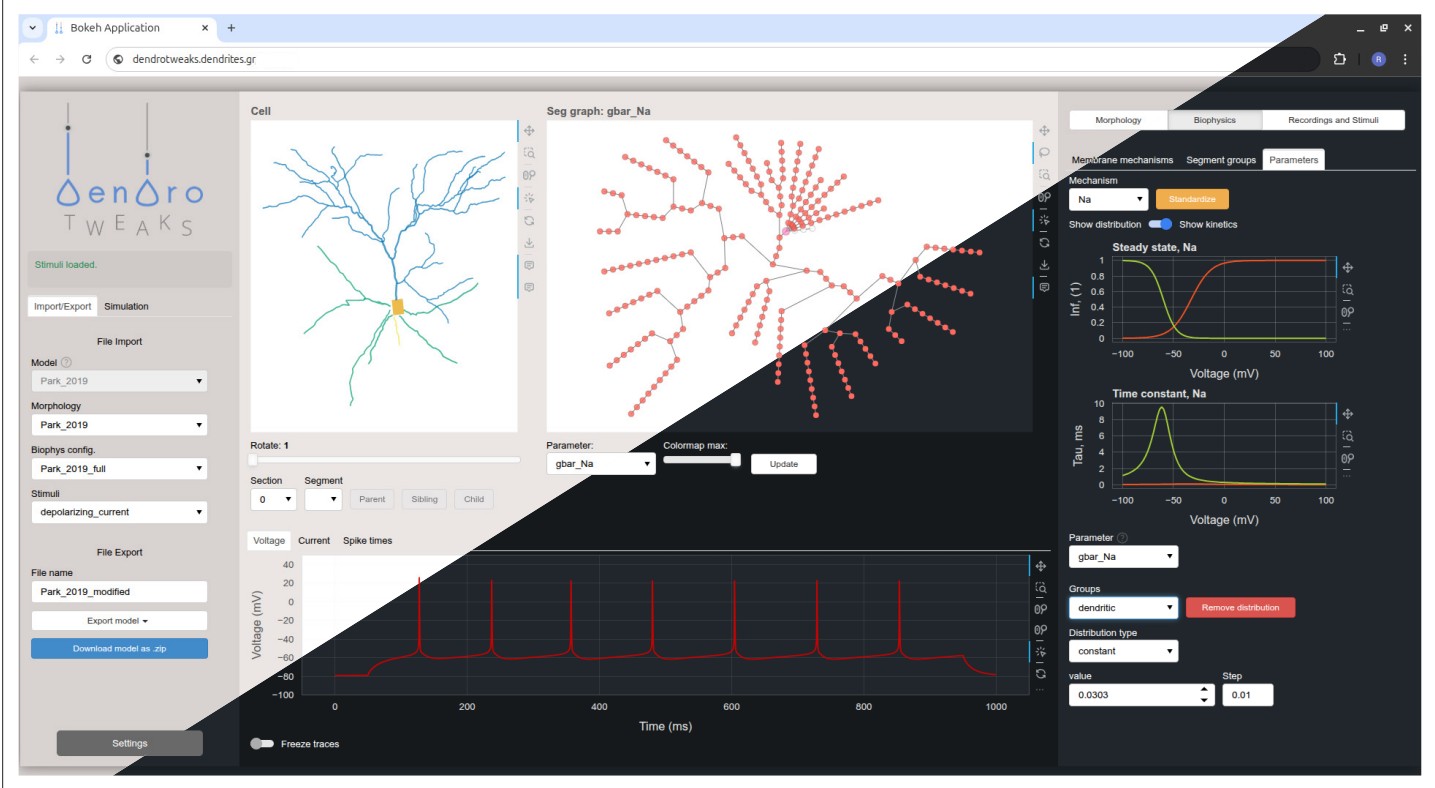

**Figure 1.** Graphical user interface (GUI). A screenshot of the web-based GUI accessed via the Chrome browser. The interface is organized into three main components: the left menu, the main workspace, and the right menu. It supports both light and dark themes.

naive to expert, to gain an in-depth understanding of the model through interactive visualization. The GUI uses the Model-View-Presenter (MVP) architecture to present a single-cell neuronal model through a web-based interface built with the Bokeh library for data visualization (**Bokeh Development Team, 2025**). The model provides a high-level simulator-agnostic representation of a cell, whereas the numerical simulation is delegated to an external simulator such as NEURON (**Hines and Carnevale, 2001**) or Jaxley (**Deistler et al., 2025**). The toolbox is available as both a standalone Python package and a web-based application. The application GUI can be accessed via the online platform (https://dendrotweaks.dendrites.gr) or a locally hosted Bokeh server. The Python package for direct interaction with the software's core functionality is distributed via the Python Package Index (PyPI; https://pypi.org/project/dendrotweaks).

The GUI is organized into three main components (**Figure 1**): (1) the left menu for file import and export operations, simulation control, and application settings, (2) the main workspace with interactive plots, and (3) the right menu with widgets and auxiliary plots. The main workspace contains top panels representing the cell and bottom panels representing monitors for the cell's activity. In the upper left corner, there is a morphology plot, where the uploaded morphology is rendered as a 2D projection of the cell. To the right of it, there is a graph representation of the cell's computational segments, where different parameters, such as channel density or synapse placement, can be visualized using a color code. The bottom panel can display time-dependent variables such as voltage, current, and input spike times. The right menu features widgets to manipulate cell morphology, spatial distributions, and kinetics of ion channels and synapses, as well as the placement and parameters of virtual recording and stimulating electrodes.

*DendroTweaks* builds a high-level modular model representation relying on commonly used neuronal data formats. It accepts neuronal morphologies in SWC format and ion channel models as MOD files. Biophysical properties of the cell are stored in JSON format, while stimulation protocols are defined through a combination of JSON and CSV files. Model configurations can be exported to this modular format at any stage and reloaded later for further use. Additionally, models developed

with *DendroTweaks* can be automatically converted into plain simulator code, enabling their smooth integration into larger network simulations.

The following sections will illustrate a potential workflow and provide a detailed description of the GUI elements and their functionality. We will begin by exploring and refining dendritic morphology and choosing the spatial discretization of the model. Next, we will explore the kinetics of ion channels and propose an automatic algorithm for standardizing such models. We will also demonstrate how different distributions of these channels across the dendritic tree can be used to reproduce dendritic phenomena, such as sodium and calcium spikes. Then, we will demonstrate how synapses can be added to our models and explore how the kinetics and placement of synapses can affect input integration within dendrites. Next, we will use an automatic morphology reduction algorithm in order to obtain a simpler model while retaining the activity close to the original one. Finally, we will present a set of experimental protocols that have been integrated into *DendroTweaks* to facilitate model validation.

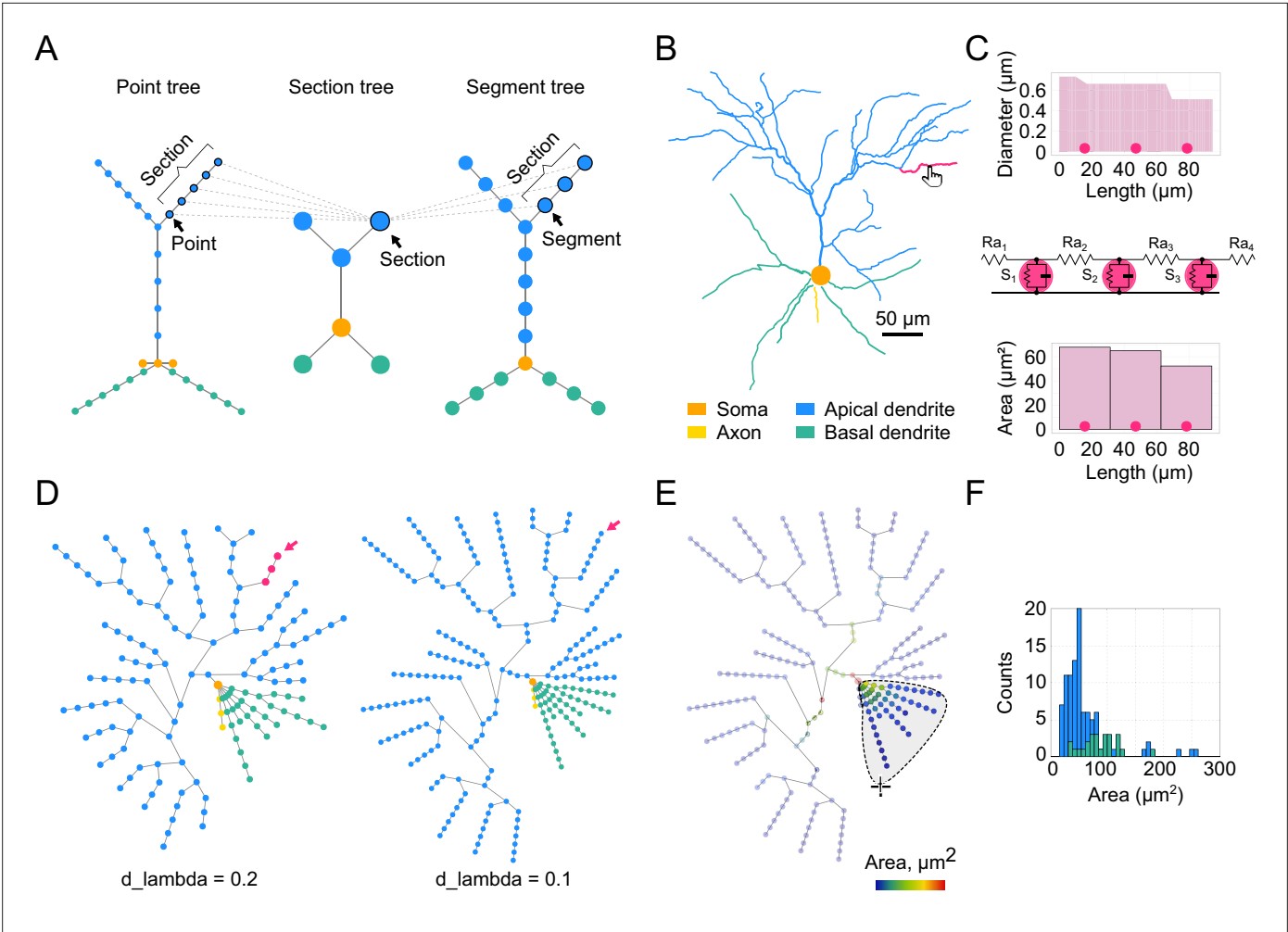

**Figure 2.** Dendritic morphology and segmentation. (**A**) A schematic illustration of a neuronal morphology as three interconnected tree graphs. Colors indicate which morphological domains the nodes belong to. (**B**) Example of an L2/3 pyramidal neuron morphology from an SWC file. The selected section is highlighted in magenta. (**C**) Detailed representation of the selected section. Top: Diameter of the selected section as a function of the section's length, with circles marking segment centers. Middle: Equivalent circuit of the selected section shown as an RC circuit, assuming a passive membrane. Bottom: Bar plot showing values of a user-selected parameter (surface area, μm²) for each segment. (**D**) Segmentation network graph representing the same cell as in (B) with d_lambda parameters of 0.2 (left) and 0.1 (right); nodes represent segments, colored as in (B**B**). (**E**) Visualization of the selected morphological parameter on the segmentation graph using a color code. The lasso mouse tool is shown, which allows the selection of specific segments. Statistical morphometric analysis can be performed for the selected part of the cell. (**F**) Histogram of segment areas for basal (green) and apical (blue) segments.

## Exploring dendritic morphology

Developing a detailed biophysical neuronal model typically begins with the experimental reconstruction of a neuron's morphology. Our toolbox accepts morphology reconstructions in the widely used SWC file format, which represents a cell as a collection of connected 3D points (nodes), where each node is defined by its `XYZ` spatial coordinates and a radius. A rich database of these files is readily available at https://neuromorpho.org (*Ascoli et al., 2007*). Moreover, online conversion from all reconstruction formats to SWC and standardization of existing SWC files is possible (*Mehta et al., 2023*). *DendroTweaks* employs tree graphs as a universal representation of neuronal morphology across different levels of abstraction: geometry, topology, and spatial discretization (*Figure 2A*). At the geometric level, nodes represent individual points in the reconstructed morphology. At the topological level, these nodes are grouped into sections, that is the parts between bifurcation points. Each section is further divided into segments, which are used in numerical simulations. These representations are linked together: each section contains pointers to its constituent points and segments, and vice versa. The toolbox features a custom module for working with such structures, enabling efficient manipulation of morphologies through operations such as insertion, removal, and translation of nodes and subtrees. Another level of abstraction is domains, that is collections of sections. A domain is a region of a neuron distinguished by its anatomical or functional properties. For example, a typical pyramidal cell has several domains: soma, axon, basal dendrites, and an apical dendrite. The apical dendrite can be further subdivided into trunk, oblique, and tuft dendrites.

Upon uploading a neuronal morphology in the GUI, it is rendered as a 2D (`XY`) projection of the cell in the main workspace on the top left `Cell` panel. This panel includes a slider for rotating the model around the Y-axis. For illustration, we use a realistic morphology of a Layer 2/3 (L2/3) pyramidal neuron of the mouse primary visual cortex (*Park et al., 2019*; *Figure 2B*). Users can navigate through the cell by selecting a section to visually inspect its parameters. This can be done by simply clicking on a section on the interactive plot or via a dropdown widget to select a specific section by its name. Navigation buttons are also available to select a parent, sibling, or child section. The parameters of the currently selected section are visualized in the right menu under the `Morphology/Section` tab with two plots (*Figure 2C*). The upper plot displays the geometry of the section, i.e., the diameter as a function of the section's length. The lower plot shows a morphological or biophysical parameter selected by the user.

Computer simulations always involve approximating a continuous system as one that is discrete in space and time, which is also applied in neuronal modeling (*Carnevale and Hines, 2006*). Discretization in time is regulated by the simulation time step, `dt`. Spatial discretization is achieved through segmentation. Each segment can be considered as an equivalent RC circuit representing a part of the membrane, with an associated set of differential equations to calculate voltage dynamics at a given point in space and time (*Figure 2C*, middle). In our example section, the centers of each segment are shown as three circles (nseg = 3), equally distributed along the section's length according to the formula $(2i - 1)/2 \times \text{nseg}$, where $i$ is an integer in the range $[1, \text{nseg}]$. The bottom panel (*Figure 2C*, bottom) features a bar plot showing the value for a chosen parameter (i.e. a NEURON range variable) for each segment. We also introduce a `Graph` view with a tree graph representing all the segments of the given cell (*Figure 2D*). This view serves three main purposes: (1) to visualize the distribution of different parameters along the dendritic tree using color code, (2) to select specific segments and update their parameters, and (3) to calculate statistics for a selected group of segments. By default, the graph plot uses color code to show the morphological domain to which a segment belongs (same colors as in the `Cell` plot). Rendering other parameters on the graph is discussed in the following sections. Since all parameters are ultimately set at the segment level, the segment graph offers the most detailed and accurate representation of the model. The granularity of the graph depends on the number of segments. In addition to defining the number of segments (nseg) for each individual section, it is possible to select the `d_lambda` parameter in the left menu that automatically assigns the number of segments for each section based on the fraction of the electrical length constant computed at the frequency of 100 Hz (*Hines and Carnevale, 2001*). In the present example for the same neuron, we demonstrate a graph obtained with a `d_lambda` value of 0.2 (*Figure 2D*, left) and with the default value of 0.1 (*Figure 2C*, right). Note that segmentation using the `d_lambda` parameter also takes into account passive cable properties `cm` and `Ra`. Finally, it is worth mentioning that the spatial discretization defined here focuses exclusively on electrical properties, whereas chemical processes operate on

fundamentally different spatial and temporal scales (*Zador and Koch, 1994*) and may require separate considerations when modeling phenomena like synaptic plasticity.

*DendroTweaks* supports basic morphometric analysis. Statistical analysis applies to an arbitrary subset of segments selected by the user on the interactive plot (*Figure 2E*). This analysis includes calculating the number of sections, segments, and bifurcations, along with average diameter, length, and area, total surface area, and total length. The number of root and leaf dendrites is displayed independently of the selection. An auxiliary histogram plot (*Figure 2F*) further aids in visualizing the distribution of any calculated parameter.

## Standardization and tuning of ion channel models

In this section, we will start exploring the biophysical properties of our model at the individual ion channel level. Ion channels are crucial in shaping dendritic and somatic voltage dynamics, which are essential for neuronal communication and information processing. Since Hodgkin and Huxley's seminal work (*Hodgkin and Huxley, 1952*), mathematical models have been vital in understanding channel kinetics (*Petousakis et al., 2023a*). The most widely used ion channel models today are written in the NMODL domain-specific language developed for the NEURON simulator (*Hines and Carnevale,*

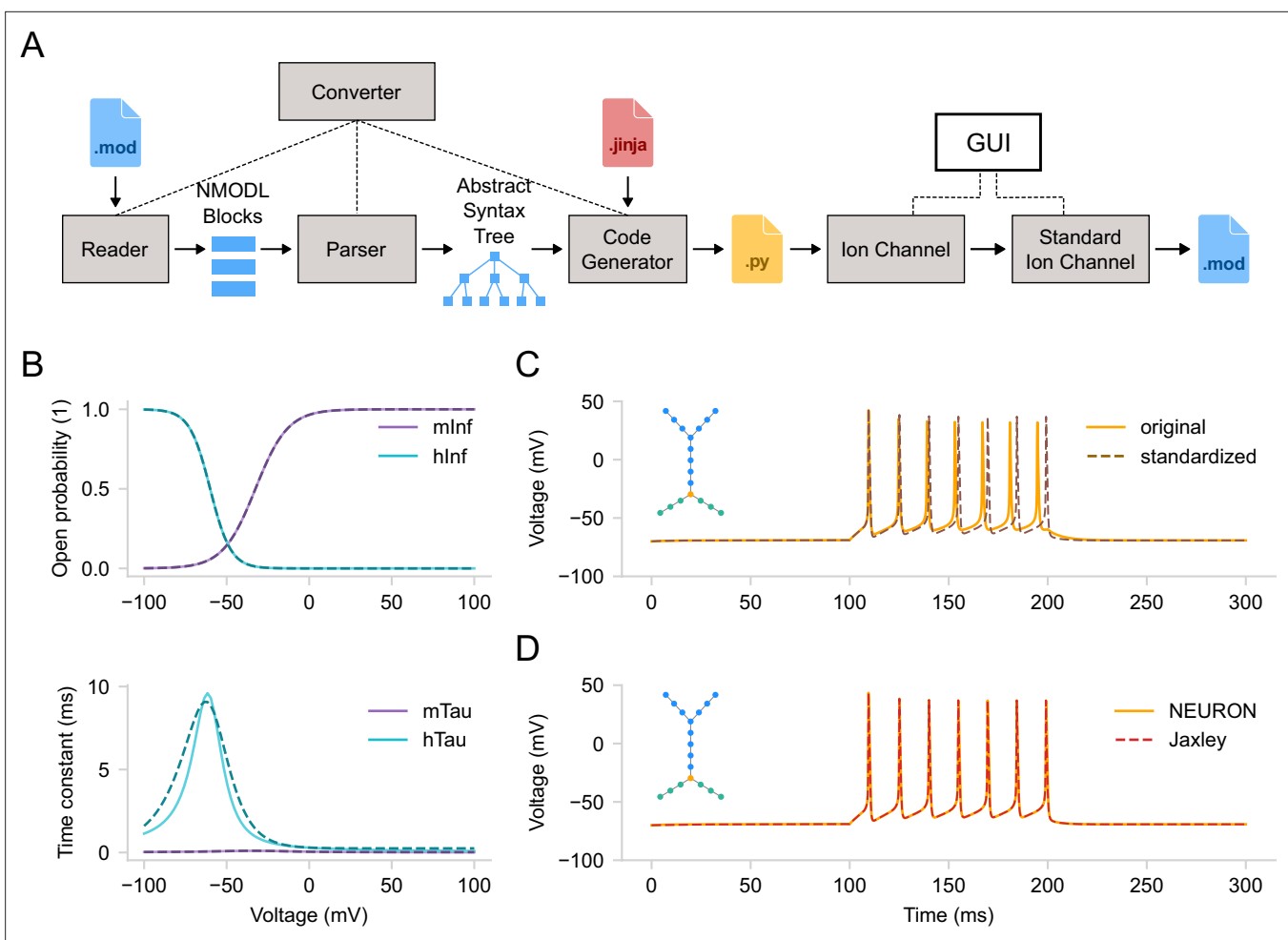

**Figure 3.** Standardization of ion channel models. (**A**) Schematic of parsing and standardizing ion channel models from MOD files. The converter automatically extracts the information from a MOD file via parsing and generates a Python file containing an `IonChannel` class. An instance of this class can be used to visualize channel kinetics. Standardization algorithm produces a `StandardIonChannel` class instance. The standardized channel model can then be exported to a new MOD file. (**B**) Kinetics of a voltage-gated sodium channel. Activation (purple) and inactivation (teal) curves for the steady-state value (top) and the time constant (bottom). The solid lines represent the original model, while the dashed lines depict the model with standardized equations fitted to the original curves. (**C**) Corresponding somatic voltage traces from the original model (solid) and one with both sodium and potassium channels standardized (dashed). Inset - the segment graph of the model. (**D**) Voltage traces for the same model (with standardized channels) constructed in DendroTweaks and simulated either in NEURON (solid) or in Jaxley (dashed).

*2000*). *DendroTweaks* features a comprehensive NMODL-to-Python converter with a custom parser written in PyParsing (https://pyparsing-docs.readthedocs.io) and automatic Python code generation using Jinja templates (*Figure 3A*). Upon importing a MOD file using the GUI, the kinetics of the corresponding channel can be visualized in the `Biophys/Parameters` tab of the right menu (*Figure 3B*). The top plot shows the voltage-dependent steady-state values of the channel's gating variables, while the bottom plot shows the corresponding voltage-dependent time constants. When a channel is selected, interactive widgets for each parameter of the channel model appear in the right menu, allowing for the dynamic update of the plots.

Manual interaction with model parameters via the GUI is informative and provides good intuition about how each parameter affects voltage dynamics. However, it does not overcome the limitations of existing models. While valid and proven useful for exploring neuronal biophysics, many existing channel models exhibit inconsistencies and deviations from theoretical formulations, limiting their interpretability and reusability. Common issues include ambiguous variable names, inconsistent equations, hardcoded parameters, lack of units, incompatibility with the latest NEURON versions (such as problems with `VERBATIM` statements and `dt`), and potential overfitting to experimental data. These issues result in a steep learning curve for inexperienced modelers and significant hurdles for experienced ones, often resulting in a lack of proper exploratory analysis, laborious manual tuning of models, and simulation errors.

Several efforts have been made to ensure more efficient and consistent channel models. In 2010, *Gleeson et al., 2010* proposed standardizing models through NeuroML, an XML-based neuronal model description language, and manually converted models of voltage- and ligand-gated conductances using the ChannelML module. In 2017, *Podlaski et al., 2017* created a framework for the automated large-scale classification of ion channels, leading to the ICGenealogy web database (https://icg.neurotheory.ox.ac.uk), which categorizes new and existing models and experimental recordings. In 2020, *Kumbhar et al., 2020* introduced a framework that parses existing MOD files to generate optimized code, significantly improving simulation speeds. Recently, a comprehensive database of the voltage-gated potassium channel (Kv) family has been made available through Channelpedia (https://channelpedia.epfl.ch) and extended to include other channel types in the Channelome project (*Ranjan et al., 2011*; *Ranjan et al., 2024*). However, none of these approaches offered an automatic, visual-guided standardization of existing MOD files through the standardization of model equations. Here, we utilize a standardization approach for MOD files using a set of equations grounded in transition state theory (see Materials and methods for details).

To demonstrate the standardization procedure, we parsed and inserted voltage-gated sodium and potassium channels and a leak channel into the cell's membrane. We then ran a simulation for 300 ms with a step current injection (amplitude 0.15 nA, delay 100 ms, duration 100 ms, temperature 37 °C). Note that the parameters of the step-and-hold stimulation protocol (i.e. amplitude, delay, duration) can be adjusted using widgets in the `Recordings and Stimuli/IClamps` tab. For standardization, we selected the sodium channel, which has activation ($m$) and inactivation ($h$) state variables (*Figure 3B*), as well as the potassium channel, which has an activation ($n$) state variable (not shown). The standardization algorithm produces plots of the original (solid) and fitted (dashed) curves for steady-state values and time constants of the state variables. Each state variable in the standardized model has five parameters: `v_half`, `sigma`, `k`, `delta`, and `tau_0` (see Materials and methods for details). Besides the activation curves, users can compare cell activity before and after standardization (*Figure 3C*). To achieve this, there is an option to 'freeze' the original current and voltage traces for comparison before running the standardization algorithm.

Although NEURON is the default simulator used in *DendroTweaks* and it was employed throughout this study, the toolbox is designed to be adaptable to other simulators. As a demonstration, we applied the same modeling interface to the recently introduced Jaxley simulator (*Deistler et al., 2025*). A key advantage of this process was that *DendroTweaks*'s NMODL-to-Python converter significantly reduced the need to manually re-implement existing ion channel models for Jaxley. By modifying the Jinja template, we were able to generate Jaxley-compatible classes that supported full numerical simulations, in addition to visualization of the channel's kinetic properties. *DendroTweaks* is capable of running simulations of multicompartmental models with multiple ion channels in Jaxley. To validate this, we ran simulations of the same simplified multicompartmental model, with sodium, potassium, and leak channels, in both NEURON and Jaxley (see *Figure 3D*). These results highlight

that *DendroTweaks* offers a unified modeling interface that can be integrated with simulators beyond NEURON in order to benefit from the additional functionality they provide.

## Distributing ion channels

Having explored and refined the neuronal morphology and individual ion channels, a user can proceed to set up the biophysical properties of their neuronal model, including the densities and distributions of the various ion channel conductances. Dendrites of most neuron types are known to express some types of active ion channels (*Johnston and Narayanan, 2008*), including major ion channel families such as Nav, Cav, Kv, KCa, and HCN (*Yu et al., 2005*). The distribution of each channel type is cell type-specific and is often non-uniform. For example, CaV1.x channels, also known as high-voltage-activated L-type channels, are densely populated in proximal dendrites (*Reuveni et al., 1993*; *Westenbroek et al., 1990*). In contrast, CaV3.x channels, or low-voltage activated T-type channels, increase in density with distance from the soma (*Magee and Johnston, 1995*; *Magee et al., 1995*), forming 'hot-spots' for dendritic Ca$^{2+}$ spikes in the apical dendrite (*Reuveni et al., 1993*; *Yuste et al., 1994*). Another compelling example is the hyperpolarization-activated mixed cation current (Ih), mediated by HCN channels, which are higher in density in distal apical dendrites of CA1 (*Magee, 1998*; *Lörincz et al., 2002*) and cortical Layer 5 (L5) (*Kole et al., 2006*) pyramidal neurons. Even passive conductances in dendrites can show non-uniform distribution (*Stuart and Spruston, 1998*). Additionally, certain kinetic properties of channels, such as half-maximal voltage, can also vary with distance from the soma (*Hoffman et al., 1997*; *Migliore et al., 1999*; *Poolos et al., 2002*). While the distribution of dendritic ion channels has been the subject of significant research, there is still much to learn about their role in neuronal integrative function (*Migliore and Shepherd, 2002*; *Johnston and Narayanan, 2008*; *Nusser, 2009*; *Shah et al., 2010*). Towards this goal, it is crucial to develop convenient tools for organizing ion channel distribution when designing software for studying neurons.

To make the process of exploring and adjusting the distributions of membrane mechanisms more user-friendly and intuitive, we utilize the graph view introduced earlier. Every membrane mechanism (distributed mechanism in NEURON) can be visualized on the graph, such that its value for a given segment is color-coded. To distribute parameters across the cell, users need to specify *where* and *how* a given parameter will be distributed. To select the segments *where* a given distribution will be applied, one needs to use segment groups. A segment group is a collection of segments that meet certain criteria (*Figure 4A*). To create a new `SegmentGroup`, one must specify both a criterion and the domains where to search for matching segments. The criterion can be one of the following types: diameter, absolute distance (to the root of the tree), or relative distance within a domain. To define *how* the parameter will be distributed, one must use distribution functions. A distribution function takes a segment's distance from the soma as input and returns the parameter value at that distance (*Figure 4B*). There are several built-in types of distributions (i.e. constant, linear, exponential, sigmoidal, etc.) which, when combined, can cover most of the existing models. Parameters of a given distribution are controlled by the widgets on the `Biophys/Parameters` tab. Note that, unlike domains, segment groups can overlap, allowing segments to belong to multiple groups simultaneously and therefore can be thought of as layers. In other words, the order of groups is important: the parameters will be assigned only from the top-most group that the segment belongs to. Moreover, a segment group can encompass multiple domains or divide sections, such that different segments within the same section may belong to different groups (*Figure 4C*).

The biophysical configuration can be exported in a JSON format that follows the structure described above. To keep the representation both compact and capable of representing complex spatial distributions, model parameters are stored using a unified, domain-based approach, where each distribution can be described with only a few coefficients (see Materials and methods for details). As a result, the same biophysical configuration can be easily applied across multiple morphologies that have the same domains.

To illustrate the effectiveness of this interface in distributing dendritic mechanisms and investigating their impact on neuronal activity, we replicated several established models. First, we reproduced the L2/3 pyramidal neuronal model from *Park et al., 2019*; *Figure 4D*, to show dendritic sodium-driven backpropagation-activated action potentials (BAPs, *Figure 4G and J*, left). Using this model, we were able to demonstrate that the distribution of sodium channels affects dendritic BAPs, as removing sodium channels from a specific branch prevented spike generation in that branch (*Figure 4J*, right).

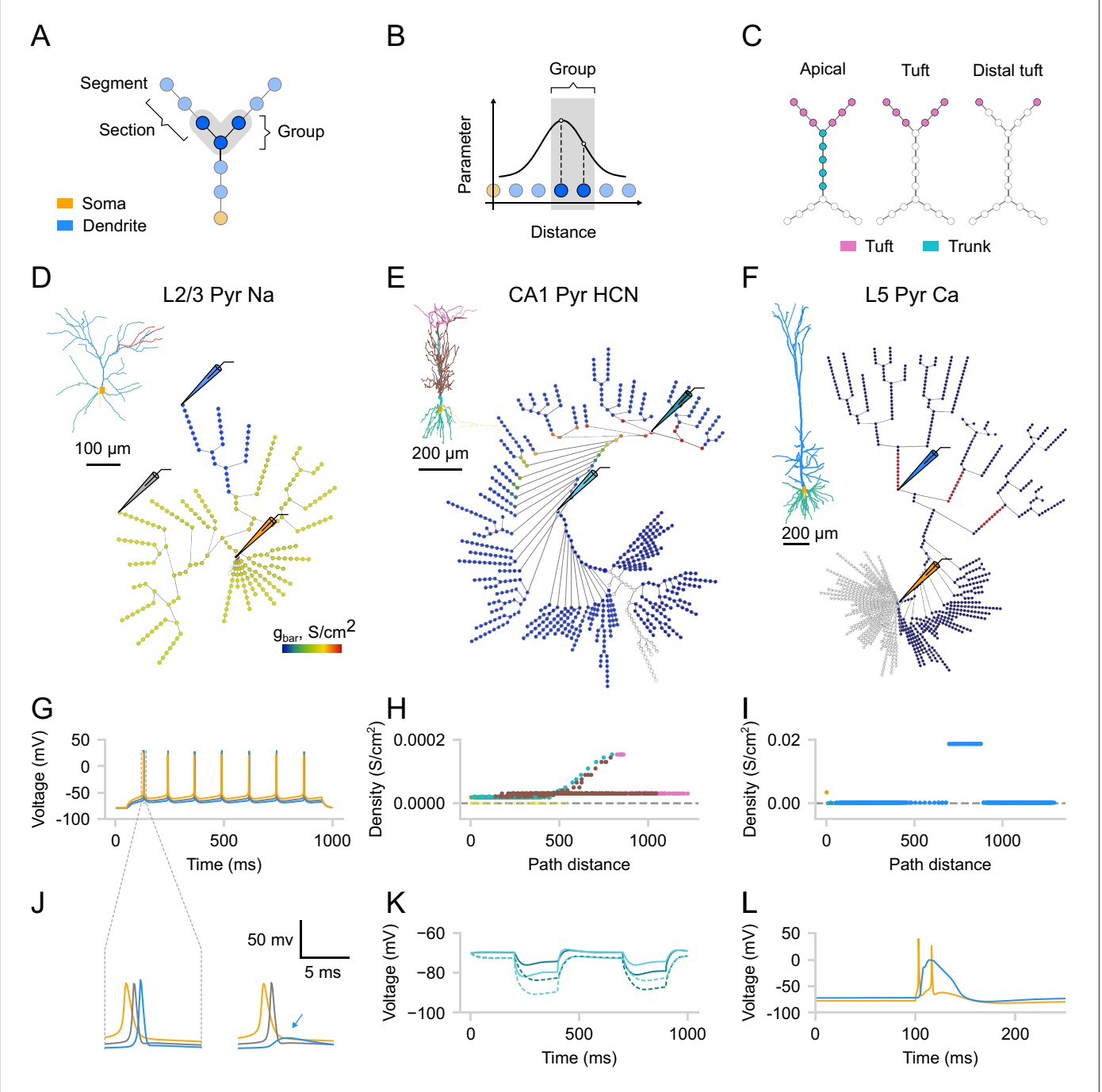

**Figure 4.** Distributions of ion channels. (**A**) Schematic showing a segment group in a ball-and-stick model, satisfying specific distance criteria. (**B**) Schematic showing distribution as a function of distance from the soma; values are assigned only to segments matching the group's criteria. (**C**) Schematic showing the difference between domains and segment groups. A group can span multiple domains (left), match a single domain (middle), or partially include individual sections (right). (**D**) Example of a constant distribution for sodium channels, where maximal conductance in the selected region (dark blue) was decreased by 60%. Schematic electrodes indicate recording positions (inset: original morphology *Park et al., 2019*). (**E**) Example of an exponential distribution for the HCN channels (inset: original morphology *Poirazi et al., 2003*). (**F**) Example of a calcium "hot spot" (red) (inset: original morphology *Hay et al., 2011*). (**G**) Sodium-driven backpropagation-activated action potentials (BAPs) evoked by a 0.162 nA somatic current injection. (**H**) Distribution of maximal HCN channel conductance as a function of distance from the soma (see functional effect in K). (**I**) Distribution of maximal calcium channel conductance as a function of distance from the soma (see functional effect in L). (**J**) Expanded time scale for the two scenarios in (D), showing failure of BAP initiation (blue arrow) in the region with reduced sodium conductance. (**K**) Voltage sag produced by HCN channels. A current step (–0.2 nA, 200 ms) is injected proximally (light cyan) and, after 300 ms, distally (dark cyan) in the apical trunk. Dashed traces: blocking HCN

*Figure 4 continued on next page*

*Figure 4 continued*

channels, modeled as 80% reduction in channel conductance. (**L**) Dendritic calcium plateau potential triggered by synaptic input at the calcium 'hot-spot' coincident with somatic current injection, leading to somatic bursting. Somatic traces are shown in orange, dendritic—in blue, cyan, and gray.

Next, we replicated the CA1 pyramidal neuronal model from *Poirazi et al., 2003* to demonstrate the effect of the experimentally observed exponential distribution of HCN channels (*Figure 4E and H*). As observed in the original study, we noted a significant depolarizing voltage sag, which was eliminated by blocking HCN channels (*Figure 4K*). Finally, we replicated the model of an L5 pyramidal neuron from *Hay et al., 2011* to demonstrate dendritic $Ca^{2+}$ spikes. We were able to evoke a dendritic $Ca^{2+}$ plateau potential by providing coincident inputs at the soma and the $Ca^{2+}$ 'hot-spot' of the apical dendrite (*Figure 4F and I*). This plateau potential then propagated to the soma, resulting in somatic burst firing (*Figure 4L*). Overall, by replicating these models and respective simulations, we demonstrate how *DendroTweaks* can aid in investigating the role of ion channels in generating dendritic events through interactive parameter adjustment and visualization.

## Distributing synapses

All previous examples used simple stimuli such as somatic and dendritic step current injections to demonstrate the functionalities of *DendroTweaks*. In this section, we describe how more realistic synaptic stimulation protocols can be implemented. Synaptic placement and timing play an important role in the integration of synaptic inputs. For example, neocortical pyramidal neurons respond supra-linearly to spatially clustered inputs and sublinearly to randomly distributed ones (*Polsky et al., 2004*; *Losonczy and Magee, 2006*; *Takahashi et al., 2012*; *Poirazi et al., 2003*; *Tran-Van-Minh et al., 2015*). Another layer of complexity to synaptic integration is added by the interplay between excitatory and inhibitory inputs (*Doron et al., 2017*; *Du et al., 2017*). Interestingly, connections from different types of inhibitory interneurons target different dendritic domains of pyramidal neurons (*Markram et al., 2004*; *van Versendaal and Levelt, 2016*), allowing for the selective regulation of information streams within a neuron. In light of the above, the ability to reproduce various synaptic input patterns is crucial for understanding dendritic integration.

As we do for ion channels, we use the graph view to visualize and allocate groups of synaptic inputs. While the pre-synaptic inputs are not modeled explicitly, we can create populations of 'virtual' neurons projecting to our model. To create a `Population` in the GUI, users must select segments (*Figure 5A*) and define population parameters using the dedicated widgets in the right menu. The creation of a population requires users to select a synapse type and specify the number of synapses to be uniformly distributed randomly within the selected segments. *DendroTweaks* offers three built-in synapse types: AMPA, NMDA, and GABA$_A$, with an additional option for a combined AMPA-NMDA synapse. Users can create as many populations as necessary. Each population can have unique kinetic parameters for the synapses (maximal conductance, `g_max`, equilibrium potential, `e`, time constants `tau_rise`, `tau_decay`) as well as parameters for incoming inputs (input rate, randomness/noise, onset, duration).

To demonstrate the power of *DendroTweaks* for exploring dendritic integration of synaptic inputs, we conducted in silico experiments involving different placements and activation times of synaptic inputs. First, we examined the effect of input synchronicity and NMDA synapses on generating NMDA spikes. Using the graph view, we distributed 20 excitatory AMPA-NMDA synapses within a single section of the *Hay et al., 2011* passive model (*Hay et al., 2011*; *Figure 5A*). We observed that simultaneous activation of synapses or activation with a Poisson spike train produced distinct dendritic voltage responses. Additionally, by blocking NMDA conductances, we were able to eliminate NMDA spikes (*Figure 5B*). Second, we replicated the effect of inhibition on dendritic NMDA spike generation, as shown in *Doron et al., 2017*. We added one inhibitory GABA$_A$ synapse in the same section and varied its activation time or location (*Figure 5C*). Consistent with the original study, NMDA spikes could not be recovered if inhibition occurred 20 ms after excitation. Moreover, proximal inhibition had little effect on NMDA spikes, whereas distal inhibition significantly reduced them. Finally, we conducted an experiment similar to that described in *Poirazi et al., 2003*, using the CA1 pyramidal neuronal model with active dendritic mechanisms. Using the graph view, we distributed 40 excitatory AMPA-NMDA synapses in two scenarios: either spread across multiple terminal branches of the apical dendrite (*Figure 5D*) or clustered within five randomly selected branches of the apical

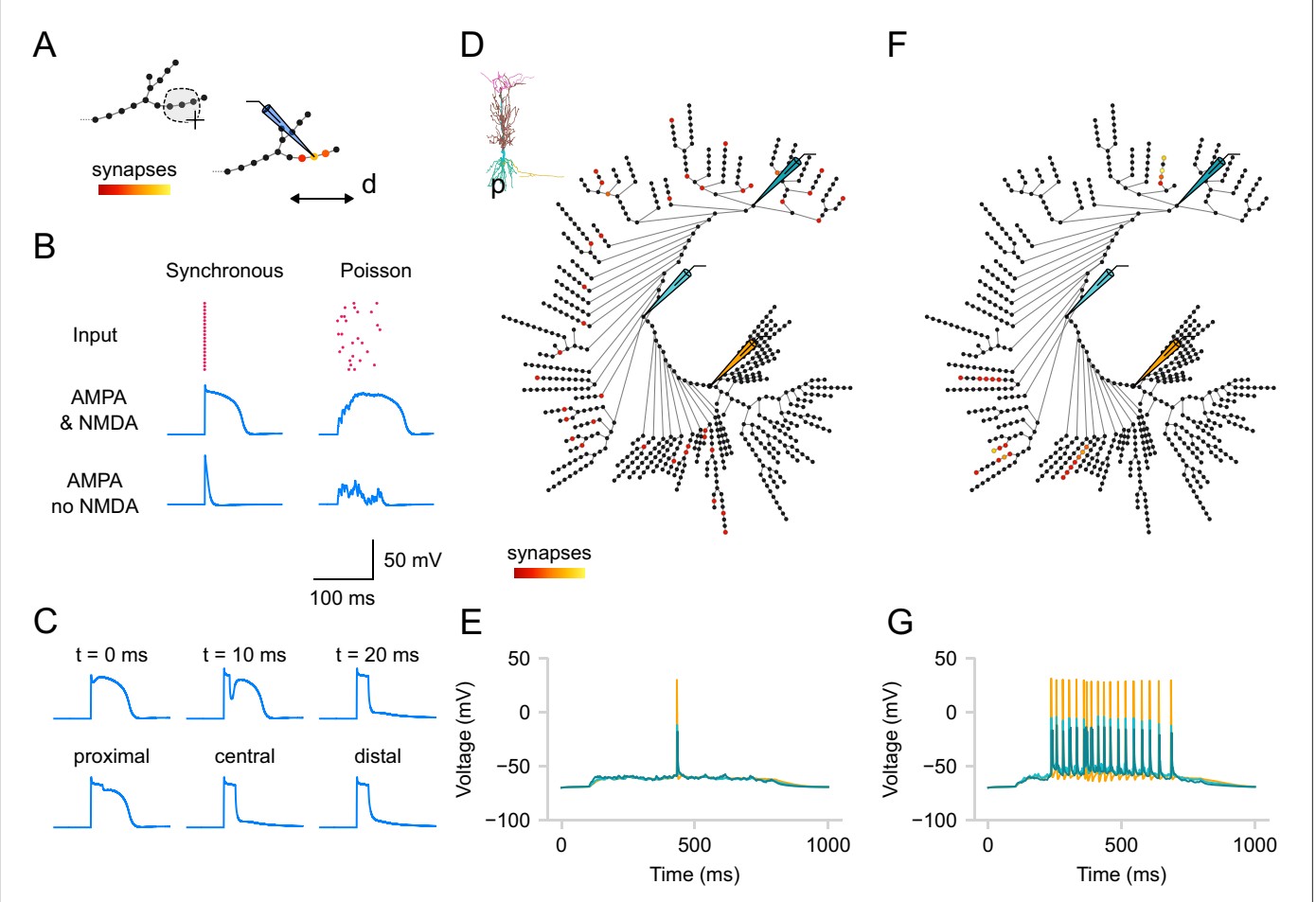

**Figure 5.** Kinetics and distribution of synapses. (**A**) Schematic representation of distributing synaptic inputs. Three central segments of a distal apical branch are selected using the lasso tool, and synapses are added (p—proximal, d—distal). (**B**) Example responses evoked by activating 20 excitatory synapses placed within one branch as in (A). The regularity of inputs varies from synchronous activation to a random Poisson spike train. Note that the raster plot for input times is accessible in one of the workspace tabs. The examples demonstrate dendritic voltage responses in the presence or absence of NMDA conductances. (**C**) Experiment similar to *Doron et al., 2017*, demonstrating the effect of inhibiting NMDA spikes. Top: One inhibitory GABA$_A$ synapse is placed in the middle of the section, and its activation time varies as 0, 10, and 20 ms after excitatory synapse activation. Bottom: The synapse location varies from the most proximal to the most distal segment of the section, with the activation time kept at 20 ms. The same stimulation protocol as in (B) with synchronous activation is used for the excitatory inputs. Scale is the same as in (B). (**D**) Distributed placement of 40 excitatory AMPA-NMDA synapses across the dendritic tree, similar to *Poirazi et al., 2003* (inset: original morphology). (**E**) Somatic and dendritic voltage responses to distributed synaptic inputs (D) (25 Hz, Poisson-distributed), which nearly fail to evoke somatic action potentials. Compare with (G) for clustered inputs. (**F**) Clustered placement of the same 40 excitatory synapses from (D) within five randomly selected branches. (**G**) Somatic and dendritic voltage responses to clustered synaptic inputs (F), demonstrating robust somatic firing activity. Somatic traces are shown in orange, dendritic - in blue and cyan.

dendrite (*Figure 5F*). As in the original study, we found that distributed synapses failed to evoke high-frequency somatic activity (*Figure 5E*), whereas clustering synapses made high-frequency somatic activity possible (*Figure 5G*). These experiments illustrate how *DendroTweaks* can facilitate the investigation of synaptic input integration and its effects at the dendritic and somatic levels.

## Reducing morphology

The explorations described in the previous sections aimed to enhance the user's understanding of which dendritic properties are essential for specific neuronal input-output transformations. With this knowledge, one can then proceed to simplify models using a built-in morphology reduction algorithm and export them for further use in faster and more interpretable neuronal networks.

Here, we follow the analytical impedance-based approach proposed by *Amsalem et al., 2020* (`neuron_reduce`). This method maps a detailed dendritic tree to an equivalent cylinder with the same

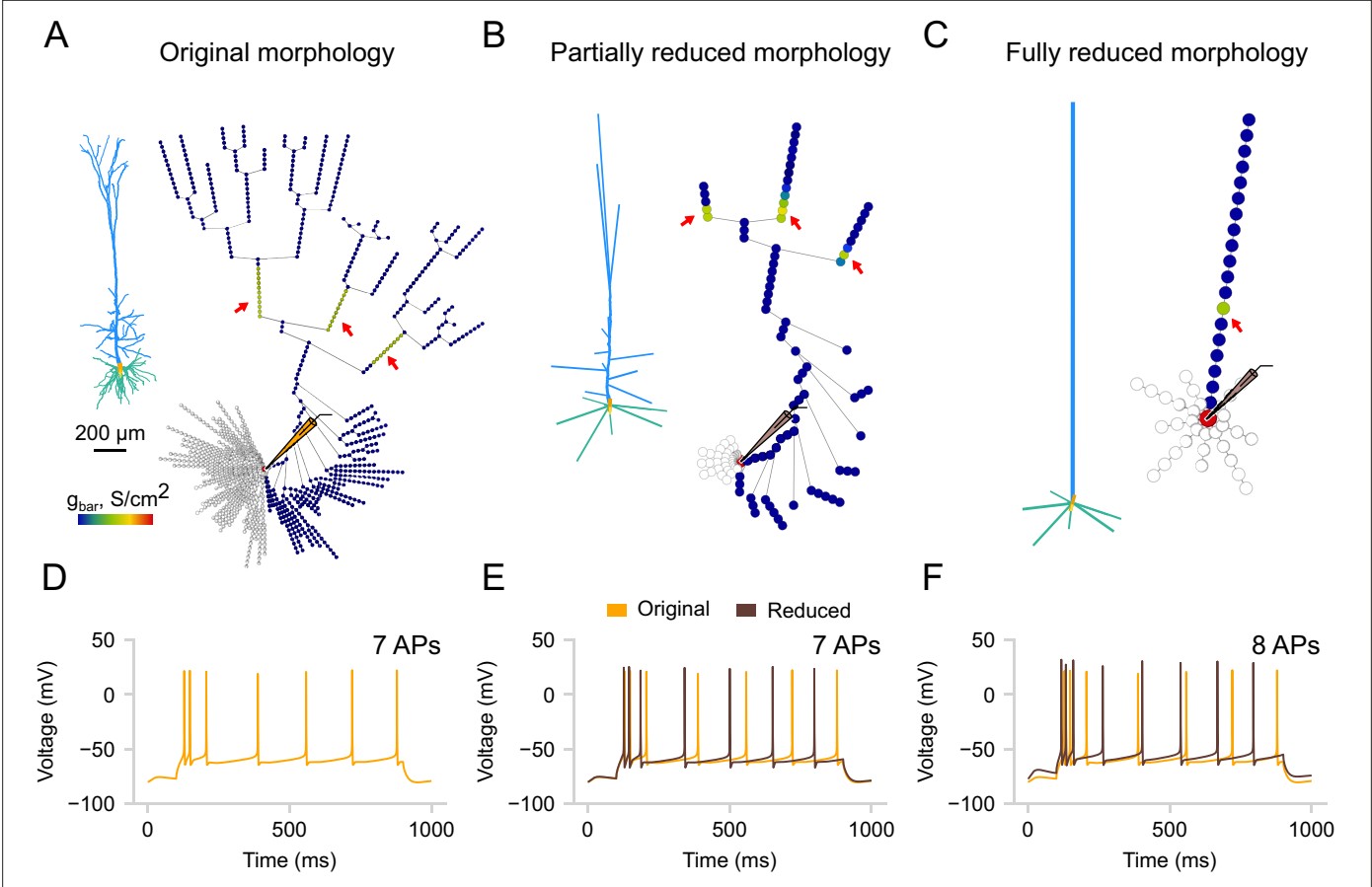

**Figure 6.** Morphology reduction. (**A**) Original morphology of L5 pyramidal neuron (*Hay et al., 2011*) and its segmentation graph showing the distribution of calcium channels. The red arrows indicate the "hot spots" with increased channel density. (**B**) Partially reduced morphology using the extended version of `neuron_reduce`. The extended version allows for the reduction of any selected branch, enabling the retention of more apical branches, in contrast to (C). (**C**) Fully reduced morphology. All stem dendrites (children of the soma) are reduced to a single equivalent cylinder. (**D–F**) Voltage response of the three models to somatic current injection of 0.5 nA. Note the difference in the number of dendritic 'hot-spots' and somatic APs between the three variations of the model. The partially reduced model more accurately reproduces the channel distribution and voltage response of the original model compared to the fully reduced one.

passive properties (i.e. specific membrane resistivity, capacitance, and axial resistivity). The transfer impedance from the distal, sealed end to the soma in the simplified model matches the transfer impedance from the most distal dendritic tip to the soma in the detailed model. Additionally, the input impedance at the proximal end matches that of the respective detailed dendrite when decoupled from the soma. In the original implementation, the entire subtree of each stem dendrite (e.g. the entire apical subtree) is mapped to a single corresponding cylinder. This, however, can impose some limitations on accurately capturing the complex branching patterns and electrotonic properties of the dendritic tree, potentially affecting the precision of simulations of synaptic integration and signal propagation.

In *DendroTweaks*, we extended the functionality of `neuron_reduce` to allow for a continuum of morphology reduction levels, bridging detailed and 'ball-and-stick'-like models. Users can select any section of the cell and reduce its subtree, which allows for any intermediate level of detail to be achieved. As an example, we start with the *Hay et al., 2011* model used in the original study (*Figure 6A*). We reduce its morphology using the original algorithm to the 'ball-and-stick' level (*Figure 6C*) and to an intermediate level where some apical oblique and tuft dendrites are preserved (*Figure 6B*). Notably, the response of the partially reduced model (*Figure 6E*) is closer to the original model's response (*Figure 6D*) in terms of the number of spikes compared to the response of the fully reduced model (*Figure 6F*). The reduction algorithm operates on passive morphologies and assigns active channel distributions *post hoc* by averaging the values of all original segments that map to a

given segment of the reduced model. To enable the export of reduced models in *DendroTweaks*'s modular format, as well as in plain simulator code, we employed fitting the parameters of a distribution function to the resulting distribution of values. This approach provided a compact representation, requiring only a few parameters to be stored in order to reproduce a distribution. To demonstrate how these exported models can be integrated into larger simulations, we implemented a 'toy' network model outside the toolbox using the reduced neurons (available in the examples directory of the package).

By integrating an enhanced version of `neuron_reduce` into *DendroTweaks*, we ensure easier post-reduction fine-tuning of model parameters. The graph view allows users to visualize the resulting distributions of channels and synapses after they have been mapped onto the reduced morphology. The simplified model can be re-validated to ensure it faithfully reproduces experimental observations. In the next section, we will discuss several built-in validation protocols that can be used for both original and simplified models.

## Validating biophysical properties

Thus far, we have presented a comprehensive set of tools available in *DendroTweaks* for developing and exploring the parameters of multicompartmental single-cell biophysical models. In addition to these functionalities, *DendroTweaks* also offers some built-in validation protocols that allow users to ensure the resulting models align with experimental observations. This approach is semi-automated, requiring users to manually implement a stimulation protocol by setting the stimulation parameters. For example, to validate somatic action potentials, a user must apply a positive step current injection at the soma to produce somatic firing. On the `Recordings and Stimuli/Analysis` tab of the right menu, selecting the `Somatic spikes` command will automatically detect action potentials and measure their properties, such as firing rate, amplitude, and half-width. While this process is not fully automated, it allows for the use of custom stimulation protocols instead of relying on predefined stimuli parameters, thereby offering more flexibility.

We demonstrate validation of passive and active properties using built-in protocols applied to the *Hay et al., 2011* model (*Hay et al., 2011*). First, we measured the input resistance (74 MΩ) and the membrane time constant (34 ms; *Figure 7A*) by applying a step current injection (–0.05 nA) at the soma while temporally blocking the HCN channels ($\bar{g}_h = 0$; *Stuart and Spruston, 1998*; *Golding et al., 2005*). We then restored the HCN channel conductance to its original values and measured voltage attenuation for either somatic (–0.5 nA, *Figure 7B*) or dendritic (–0.05 nA, *Figure 7C*) step current injection. Next, we stimulated the soma with a positive step current (0.793 nA) and detected somatic action potentials (*Figure 7D*). From the same trace, we measured the peaks, amplitudes, and half-widths of individual action potentials (*Figure 7E*). We then constructed a somatic f-I curve (*Figure 7F*) by applying current steps of increasing amplitude (0.1 nA step). After validating the somatic activity, we evaluated dendritic integration nonlinearity (*Figure 7G*, left) by comparing measured individual waveforms (*Figure 7G*, right) to the expected linear summation of postsynaptic potentials (PSPs). Note that protocols for both the somatic f-I curve and dendritic integration curve require multiple simulation runs with varying stimulus intensity, which are done automatically after the user specifies the initial stimulation parameters. Finally, since the model has HCN channels, we measured the voltage sag ratio (0.14) and the steady-state input resistance (40 MΩ) by applying a negative step current injection (–0.05 nA) at the soma (*Figure 7H*). A 300 ms period before the simulations (not shown) was used to stabilize the resting membrane potential. For a detailed description of the protocols, see *Table 1* in Materials and methods.

## Discussion
### Conceptual and implementational accessibility

We developed *DendroTweaks* to make detailed biophysical models with active dendrites more accessible at the conceptual and implementational levels. At the conceptual level, our main motivation was to illuminate how morpho-electric properties of dendrites shape neuronal activity and computations. We deliberately focused on single-cell models to provide comprehensive functionality for tuning subcellular properties, including dendritic morphology, ion channel kinetics and distributions, as well as synaptic inputs. With *DendroTweaks*' interactive interface, users can better understand

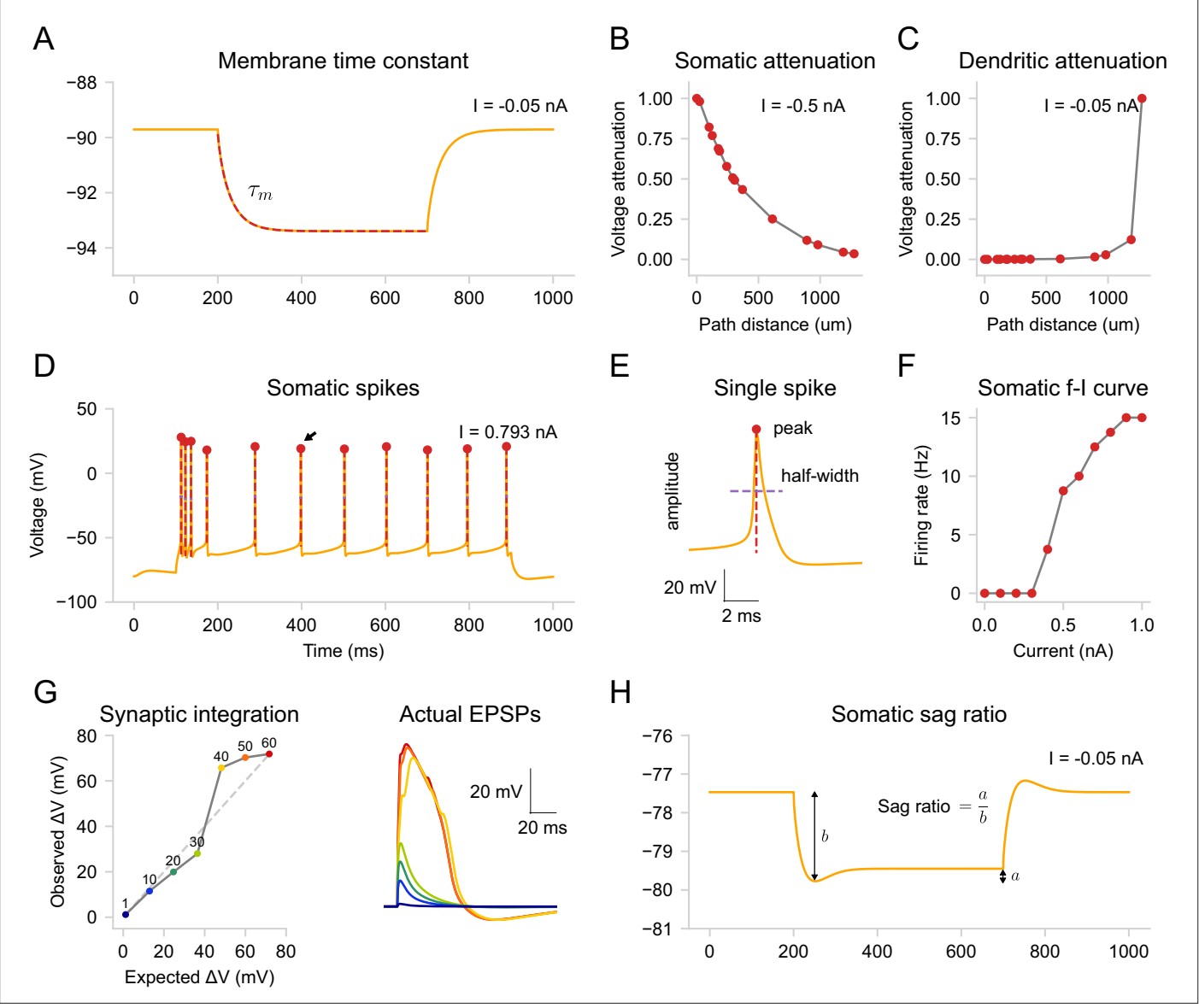

**Figure 7.** Validation protocols. Built-in validation protocols applied to the *Hay et al., 2011* model (*Hay et al., 2011*; see also *Table 1*). (**A**) Membrane time constant (34 ms) measured by applying a step current injection (–0.05 nA) at the soma, while blocking the HCN channels. (**B**) Voltage attenuation for somatic (–0.5 nA, left) and dendritic (–0.05 nA, right) step current injection at all bifurcation points along the path from a selected tip segment. (**C**) Detected somatic action potentials from stimulation with a positive step current (0.793 nA). (**D**) Single action potential indicated with an arrow in (C), with measured peak, amplitude, and half-width values. (**E**) Somatic frequency-current curve constructed by applying current steps of increasing amplitude (0.1 nA step) at the soma. (**F**) Nonlinear integration of synaptic inputs in a tuft dendrite. Left: Expected vs. actual EPSP amplitude for 1–60 synchronously activated AMPA-NMDA synapses. Right: Actual EPSP waveforms. (**G**) Voltage sag ratio at the soma measured by applying a negative step current injection (–0.05 nA). Note that, unlike in (A), the hyperpolarization-activated current through the HCN channels is present here.

how these subcellular properties influence dendritic activity and neuronal output in their models. We equip neuronal models with widgets and interactive plots, making every parameter visually accessible and interactively tunable. Importantly, for simple models, plots respond to user actions in real time, ensuring a smooth model exploration and tuning process. Interactive plots illustrating neuronal morphology significantly simplify the process of navigating through various sections and segments of the model. We enhance the interactivity of ion channel models by providing visualization of channel kinetics based on MOD files. Furthermore, by representing a cell as a graph with computational segments as nodes, we simplify and visually enhance the process of distributing ion channels and synapses throughout the cell. Finally, we extend the `neuron_reduce` approach for

**Table 1.** Validation protocols.

| Validation protocol | Readout | Minimal set of membrane parameters | Recordings | Stimuli |
|---|---|---|---|---|
| Input resistance | $R_{in}$, MΩ | cm, Ra and Leak | Any single segment | Negative step current injection |
| Membrane time constant | $\tau$, ms | cm, Ra and Leak | Any single segment | Negative step current injection |
| Rheobase current | $I_{rh}$, nA | At least Na and Kdr | Soma | Positive step current injection |
| Somatic spikes | AP times, amplitude, half-width, rate, ISI | At least Na and Kdr | Soma | Positive step current injection above the rheobase value |
| Somatic f/I curve | $\text{Rate}(I_{ext})$, $Hz$ | At least Na and Kdr | Soma | Positive step current injection (amplitude automatically increased) |
| Dendritic nonlinearity | EPSP, mV | Dendritic ion channels | Dendritic segment | Excitatory synapse (weight automatically increased) |
| Voltage attenuation | $\Delta V_0/\Delta V_i$, unitless | cm, Ra and Leak | At least two segments | Negative step current injection |
| Sag ratio | $\dfrac{V_{offset} - V_{min}}{V_{onset} - V_{min}}$ | HCN channels (h current) | Any single segment | Negative step current injection |

morphology reduction to encompass all potential reduction levels from full morphology to 'ball-and-stick'-like models and incorporate this method into our graphical interface, thereby taking advantage of inherent visualization and validation capabilities.

At the implementational level, our goal was to provide a high-level modeling framework that abstracts away technical implementation details and produces structured, transparent, and interoperable model representations. We provide users with a unified modeling interface that allows them to construct high-level simulator-independent model representations while automatically generating simulator-specific model instances. The underlying model structure is designed to be modular, clearly separating morphology, biophysical configurations, and stimulation protocols. To comprehensively capture complex dendritic properties, we introduced a custom format for biophysical parameters, capable of capturing non-uniform ion channel distributions across different dendritic domains. To enhance data management, we introduced versatile tools for parsing, generating, and standardizing commonly used neuronal data formats. In particular, we developed rich functionality for processing ion channel models written in the NMODL language, making them more comprehensible and reusable. To ensure seamless integration into broader modeling workflows, we implemented a range of export options, allowing users to reuse individual model components or entire models across different modeling software. Through this combination of advanced model construction capabilities, comprehensive data handling, and broad interoperability, *DendroTweaks* complements existing tools and fills an important niche within the neuronal modeling ecosystem. An integrated modeling workflow might include multiple steps and platforms. For example, users can first set up a model in *DendroTweaks* by removing morphological artifacts and standardizing ion channel models. They can then establish meaningful initial parameters and use external tools for automated parameter optimization. Subsequently, users can re-import the model for visual exploration, validation, or morphology reduction. Finally, they can export these refined models and use specialized software to scale them up into larger network simulations. In the following paragraphs, we compare *DendroTweaks* with various modeling software to specify its place in the neuronal modeling landscape.

## Comparison to existing modeling software

Single-cell modeling encompasses a diverse range of practices, from refining morphological data and optimizing biophysical properties to visualizing and analyzing neuronal dynamics. Over the years, numerous tools have been developed to aid in the creation, visualization, and optimization of neuronal models. Primary simulation environments like NEURON (*Hines and Carnevale, 2001*), NEST (*Gewaltig and Diesmann, 2007*), and Brian 2 *Stimberg et al., 2019* have been complemented by a variety of auxiliary software tools designed to enhance interaction with model parameters.

Several tools have been developed to assist in visualizing and editing morphological reconstructions of real neurons, such as neuTube (*Feng et al., 2015*), REMOD (*Bozelos et al., 2015*), HBP MORPHOLOGY VIEWER (*Bakker et al., 2017*), and HUGO (*Aliaga Maraver et al., 2018*). *DendroTweaks* includes a simulator-independent tree graph construction module, enabling efficient morphology manipulation through operations such as inserting, removing, and translating nodes or subtrees. Its graphical interface offers the functionality for inspecting and refining morphological parameters to identify reconstruction artifacts and 'bugs' through visual exploration and statistical analysis. Nevertheless, the toolbox lacks advanced tools, such as 3D mesh editing and neuronal growth modeling capabilities. For more extensive morphology-focused needs, users are directed to specialized software like the TREES toolbox (*Cuntz et al., 2011*), Neuronize (*Brito et al., 2013*), NeuroEditor (*Velasco et al., 2024*), and NETMORTH (*Koene et al., 2009*).

With a growing number of models available online from repositories like https://modeldb.science (*McDougal et al., 2017*), https://neuromorpho.org (*Ascoli et al., 2007*), and https://celltypes.brain-map.org (*Gouwens et al., 2018*; *Gouwens et al., 2019*), standardization is crucial for ensuring reproducibility and reusability in neuronal modeling. Existing standards like SONATA (*Dai et al., 2020*), NeuroML (*Gleeson et al., 2010*), and NineML (the *Gorchetchnikov, 2010*) provide frameworks for model description. However, manual standardization can be laborious and prone to errors. Importantly, *DendroTweaks* is not presented as a new standard but as a tool that conforms to a given standard to automatically standardize a model. Future releases might include an automatic export to an expanded range of formats, such as ChannelML (*Gleeson et al., 2010*). Finally, to our knowledge, *DendroTweaks* is the only tool that can simultaneously parse existing ion channel model files, allow for visual exploration and fine-tuning of their kinetics and distributions via a GUI, and offer automatic standardization.

Another critical aspect of neuronal modeling is optimization of biophysical parameters. Evolutionary (*Van Geit et al., 2016*), Bayesian (*Gonçalves et al., 2020*), and gradient-based methods (*Jones and Kording, 2024*; *Deistler et al., 2025*) have been proposed to offer data-driven model parameter optimization. While the recent advent of such automated optimizers opens new possibilities for large-scale modeling, we argue that manual exploration remains a valuable and complementary approach. Human oversight provides both initial parameters before optimization and essential sanity checks after, making intuitive understanding a must when validating model behavior. *DendroTweaks* perfectly complements black-box optimization by providing an intuitive environment for hands-on parameter tuning and model refinement. Incorporating automatic optimization algorithms into *DendroTweaks* alongside interactive visualizations presents a promising future direction, combining the strengths of both approaches.

A range of validation frameworks has been developed to ensure that optimized parameter sets remain biophysically plausible. These approaches typically perform systematic searches across the parameter space and explicitly quantify heterogeneity, that is populations of models that reproduce experimental observations while capturing biological variability, and degeneracy, that is the ability of different parameter combinations to yield similar outputs (*Migliore et al., 2018*; *Roy and Narayanan, 2021*; *Roy and Narayanan, 2023*; *Reva et al., 2023*). *DendroTweaks* approaches heterogeneity and degeneracy differently. Rather than generating model populations, it enables users to interactively navigate the parameter space, visualizing neuronal activity and applying built-in validation protocols to assess whether individual model configurations fall within biologically realistic regimes. While population-based frameworks excel at systematic large-scale exploration, *DendroTweaks* complements them by supporting hypothesis-driven interrogation of how specific parameters influence neuronal function.

Visualization and the development of intuitive graphical user interfaces have been crucial in neuronal modeling. One of the most successful examples is NetPyNE (*Dura-Bernal et al., 2019*), which offers a graphical interface for data-driven multiscale network modeling in NEURON. Meanwhile, *DendroTweaks* focuses on the subcellular level, providing a more explicit single-cell model interface. It facilitates an interactive approach to visualize and modify morphological parameters, ion channel kinetics and distributions, as well as to observe activity in different compartments. This capability is particularly important for models with active dendrites. We envision as a good practice exporting fine-tuned single-cell models from *DendroTweaks* and incorporating them into complex networks in NetPyNE.

Indeed, the ultimate goal of creating single-cell models is often to integrate them into a network. In this context, simplifying models becomes another crucial aspect of neuronal modeling. Morphology reduction is a common simplification technique with a long history. Pioneering works (**Stratford et al., 1989**; **Bush and Sejnowski, 1993**) aimed to conserve axial resistance, reducing multicompartmental models to 8–9 compartments. Later approaches focused on preserving voltage attenuation (**Destexhe, 2001**) or surface area (**Hendrickson et al., 2011**; **Marasco et al., 2013**). The most recent methods such as `neuron_reduce` (**Amsalem et al., 2020**) and NEAT (**Wybo et al., 2021**) provide analytical solutions to the reduction problem by preserving the impedances of the original model. In *DendroTweaks*, we adopted the `neuron_reduce` approach and extended it to support a continuum of reduction levels, allowing users to gradually reduce dendritic subtrees. This added flexibility is important because dendritic trees rely on semi-independent subunit integration, which enhances localized processing and computational complexity. Spatial organization of inputs plays a crucial role in their integration, with clustered inputs being more likely to drive somatic firing in pyramidal neurons (**Poirazi et al., 2003**; **Polsky et al., 2004**; **Losonczy and Magee, 2006**). When multiple synaptic inputs from otherwise isolated branches are remapped to a reduced cylinder, clustering can occur, substantially altering the integration process. Therefore, the intermediate levels of reduction we added to `neuron_reduce` could potentially allow for more accurate implementation of independent input integration within dendritic subunits. A compelling example of when this might be needed is recent research (**Otor et al., 2022**), which showed that early bifurcating L5 pyramidal neurons exhibit pronounced functional compartmentalization of the apical dendrite, correlating with behavioral variables.

## Limitations and further directions

*DendroTweaks* addresses a broad spectrum of single-cell modeling needs. However, it is important to acknowledge its limitations and outline promising future directions. The key feature of *DendroTweaks*, its real-time interactivity, is constrained by the performance of the underlying simulator. When running hyper-detailed simulations with numerous segments over extended periods of time, the simulation can become slow, reducing the interface's responsiveness and real-time update capabilities. To enhance performance, future versions of *DendroTweaks* should consider integrating faster simulation methods or developing a custom single-cell simulator. A promising approach is to run simulations using the optimized CoreNEURON (**Kumbhar et al., 2019**) and Dendritic Hierarchical Scheduling (DHS) algorithms (**Zhang et al., 2023**), which have been shown to greatly speed up NEURON simulations, or leverage Jaxley's built-in functionality for JIT compilation and GPU usage (**Deistler et al., 2025**).

While the current implementation of *DendroTweaks* is based on the NEURON simulator, the approach is essentially simulator-agnostic. As we have demonstrated for Jaxley, its functionality can be further extended to include other simulators. While our proof of concept shows that a multi-compartment model with multiple ion channels can be simulated using Jaxley as an alternative to NEURON, fully preserving the programming interface, this implementation is not optimized to leverage the full range of Jaxley capabilities and still lacks some of the features that will be added in future releases. Another promising future direction is to introduce an automated conversion of simplified biophysical models to integrate-and-fire few-compartmental models implemented in Brian 2 using Dendrify (**Pagkalos et al., 2023**). This would allow for the extension of the proposed workflow by automatically simplifying not only the morphology but also the biophysics of a neuron, spanning any level of conceptual granularity.

While *DendroTweaks* models rely on specific parameters for ion channel kinetics and spatial distributions, experimental measurements of these properties reflect substantial variability across cells. Averaging kinetic data or conductance gradients across multiple cells can create the misleading impression of fixed parameter values, whereas in reality, each neuron may exhibit a distinct profile. Degeneracy further complicates modeling, as multiple, widely different combinations of kinetic and distribution parameters can produce nearly identical neuronal responses, making it difficult to infer unique underlying mechanisms. While *DendroTweaks* does not resolve these issues inherent to most neuronal models, it allows users to better understand them by interactively adjusting parameters and immediately observing their effects on model outputs.

Another limitation of the current *DendroTweaks* implementation is the range of ion channel models that can be standardized. As of now, only voltage-gated channels using the Hodgkin-Huxley (HH)

formalism can be standardized. Even though Markov chain state-based kinetic models might offer a more accurate representation of ion channel kinetics (*Lampert and Korngreen, 2014*), they are not supported by *DendroTweaks*. Nevertheless, it is important to note that most models use the HH formalism, as Markov models can be more complex and slower to run. Additionally, our parsing and automated standardization algorithm relies on specific heuristics and cannot handle significant deviations in MOD file code patterns. For example, the parser assumes that variable names for the time constant will include 'tau' (case insensitive). Therefore, for some files, the algorithm may require some minor preprocessing and manual tuning by the user.

In its current implementation, *DendroTweaks* offers relatively limited capabilities for representing synaptic inputs. Synaptic parameters are defined at the population level, and setting per-synapse properties is not yet supported. Synapse placement within a user-specified region follows a uniform random distribution. In addition, activation properties are specified abstractly in terms of rates, noise levels, and durations, without support for user-defined input sequences. Extending the toolbox to include per-synapse control, synaptic scaling (*Magee and Cook, 2000*), plasticity mechanisms, and more flexible connectivity definitions would substantially broaden the range of research questions that can be addressed using *DendroTweaks*.

The extended functionality of the morphology reduction algorithm in *DendroTweaks* provides a continuum of reduction levels that better preserve properties of the original model. However, even intermediate reduction levels inevitably alter integrative properties and eventually lead to the loss of dendritic computations. Nevertheless, it is worth noting that assessing how exactly morphology reduction affects integrative properties is one of the key use cases for *DendroTweaks*, thanks to its exploratory capabilities. Importantly, we don't view reduction as an ultimate goal for any model but rather encourage approaches where the same system can be studied at different levels of granularity, as elegantly shown in *Billeh et al., 2020*. Beyond optimizing simulation speed, morphology reduction serves as a valuable technique to explore the importance of dendritic compartmentalization. The question of whether the computational unit of a neuron is a single spine, a branch, or an entire domain remains open (*Häusser and Mel, 2003*; *Francioni and Harnett, 2022*; *Stuyt et al., 2022*). Using morphology reduction at multiple levels in in silico experiments can help address this question by revealing how dendritic morphology shapes compartmentalization, which in turn determines the neuron's input-output transformation properties.

Finally, there is further potential for interoperability of *DendroTweaks* with other neuronal modeling software. *DendroTweaks* is designed to read and write the most popular formats for representing neuronal morphology (SWC) as well as ion channel models (MOD). It is possible to automatically generate plain NEURON (Python) code to export a refined model and use it outside the toolbox. Having the same option for other simulators, such as Jaxley, would also be beneficial. To enhance compatibility with existing models and the reusability of standardized models, support for more file formats and standards needs to be incorporated. The SONATA data format (*Dai et al., 2020*), designed to efficiently represent network models and simulation data, plays an important role in this regard. Integration with this format is currently under development, and support for exporting models in SONATA is planned for future releases. Another notable example is NeuroML (*Gleeson et al., 2010*), an XML-based neuronal model description language that provides a standardized notation for both morphological and biophysical parameters. By supporting such widely adopted standards, *DendroTweaks* can be more seamlessly integrated into a larger ecosystem of interoperable, open-source software (*Sinha et al., 2024*).

## Conclusion

We believe that *DendroTweaks* will be appealing to a wide range of researchers. For those new to computational modeling, it provides an intuitive understanding of how model parameters influence neuronal dynamics, making it also a valuable educational tool. It lowers conceptual and technical barriers to biophysical neuronal modeling for experimentalists and specialists from adjacent fields. Given the growing interest in dendritic computations within artificial intelligence and neuromorphic computing, *DendroTweaks* offers clear visualizations of dendritic integration, benefiting those less familiar with biophysical modeling. For experienced modelers, *DendroTweaks* offers a comprehensive workflow, from single-channel analysis to validating dendritic properties, with tools for visual inspection of cell topology, geometry, channel kinetics, and their distributions, as well as neuronal activity

under variable stimuli, facilitating visual debugging. It also includes standardization and morphology reduction tools to improve model tractability and reusability for network simulations. *DendroTweaks* evolves with community feedback, adding new features over time. More than just a tool, *DendroTweaks* is a versatile framework that can integrate other visualizations and algorithms, ensuring its lasting relevance in the research community.

## Materials and methods

### User interface

The user interface of *DendroTweaks* is implemented using Python 3, following the Model-View-Presenter (MVP) architectural pattern. The `Model` class defines a biophysical neuronal model and also serves as the core of a standalone Python package, which exposes the toolbox's main functionality for programmatic use. For the `View` class, we use the Python Bokeh library (**Bokeh Development Team, 2025**), which facilitates data visualization by generating the necessary JavaScript code to build the web-based interface. The `Presenter` is a class that acts as an intermediary, processing user commands, updating the `Model` accordingly, and ensuring that the `View` reflects the current state of the `Model`. This separation of concerns ensures a clean and maintainable codebase, allowing for efficient data handling and user interaction. The source code is openly available on GitHub for both the standalone Python package https://github.com/Poirazi-Lab/DendroTweaks (**Makarov et al., 2025a**) and the web application https://github.com/Poirazi-Lab/DendroTweaksApp (**Makarov et al., 2025b**). Detailed documentation and tutorials are available through ReadTheDocs (https://dendrotweaks.readthedocs.io/en/latest).

### Biophysical models

To demonstrate the capabilities of the toolbox, we employed three well-established biophysical neuronal models with detailed morphology. The first model is an L2/3 pyramidal neuronal model with morphology reconstruction from **Park et al., 2019** biophysical mechanisms originally developed by **Mainen and Sejnowski, 1996**, further refined by **Smith et al., 2013**, and recently utilized in **Park et al., 2019** and **Petousakis et al., 2023b**. The membrane potential was initialized at $V_{init} = -79$ mV, and simulations were conducted at 37°C. The equilibrium potentials were set as follows: $E_{Leak} = -79$ mV, $E_{Na} = 60$ mV, $E_K = -80$ mV, and $E_{Ca} = 140$ mV. The second model is a widely used L5 pyramidal neuronal model with morphology reconstruction and biophysical mechanisms from **Hay et al., 2011**. For this model, the membrane potential was initialized at either $V_{init} = -80$ mV or $V_{init} = -90$ mV, with simulations performed at 37°C. The equilibrium potentials were $E_{Leak} = -90$ mV, $E_{Na} = 50$ mV, $E_K = -85$ mV, and $E_{Ca} = 132$ mV. The third model is a CA1 hippocampal pyramidal neuron, based on **Poirazi et al., 2003** and **González, 2011**. Here, the membrane potential was initialized at $V_{init} = -70$ mV, with simulations conducted at 34°C. The equilibrium potentials were $E_{Leak} = -70$ mV, $E_{Na} = 50$ mV, $E_K = -77$ mV, and $E_{Ca} = 140$ mV. All simulations were executed on a Dell G15 5515 laptop (Ryzen 7 5800 H, 16 GB RAM, Linux Ubuntu 20.04 LTS) with a spatial discretization factor $d_\lambda = 0.1$ and a time step $dt = 0.025$ ms.

### Data format

*DendroTweaks* uses a custom modular data format tailored to single-cell modeling needs. This format builds on widely used file types. Neuronal morphology is represented in SWC files, which *DendroTweaks* can read, modify, and export using a custom graph-processing module. Ion channel models and other membrane mechanisms are described in MOD files (see the following section for details). The biophysical configuration of a cell is specified in JSON files, adapted from the Allen Cell Types Database schema (**Gouwens et al., 2018**) and extended to support non-uniform parameter distributions. Each JSON file contains three main sections: (1) the mapping of domains to inserted membrane mechanisms, (2) the definition of segment groups, and (3) the mapping of segment groups to distribution functions for each model parameter. This structure enables concise and unified representation of complex distributions for any biophysical parameter, including passive and kinetic ones. Simulation, recording, and stimulation parameters are also stored in JSON files, while element-wise parameters, such as spatial locations of recordings and stimuli, are stored in CSV files. Each model is organized within a dedicated folder containing `morphology`, `biophys`, and `stimuli` subfolders. User-defined

MOD files must be placed in the `biophys/mod` folder. Once this structure is established, the toolbox takes on most of the file management responsibilities from the user (e.g. specifying file paths, compiling MOD files, etc.).

## Morphology processing and simulator-specific models

Building a neuronal morphology from an SWC file in *DendroTweaks* is implemented independently of any specific simulator. Each model is represented by interconnected tree graphs capturing the morphology at three abstraction levels: point tree, section tree, and segment tree. This structure allows users to build the model incrementally, starting from the point tree, for more precise control over morphology refinement. Each tree encapsulates level-specific properties and provides methods for common operations, such as sorting, inserting, and removing nodes or subtrees. Nodes maintain references to their children and parent, as well as attributes describing level-specific properties. Sections have access to their points and segments, and vice versa, linking all levels together. Additional methods are available for computing path distances, which are used when assigning parameter distributions. In parallel, *DendroTweaks* automatically generates corresponding sections and segments within the selected simulator, either NEURON (*Hines et al., 2009*) or Jaxley (*Deistler et al., 2025*), and maintains the references from its own sections and segments to the simulator objects. This enables seamless retrieval and modification of simulator-specific parameters while maintaining a simulator-independent scaffold.

## NMODL to Python conversion

The `Converter` class encapsulates three components: a `Reader`, a `Parser`, and a `CodeGenerator`. The `Reader` performs basic preprocessing, such as removing comments and splitting the input into NMODL blocks, providing a structured basis for efficient parsing and handling of individual components. The `Parser`, implemented with the PyParsing library (https://pyparsing-docs.readthedocs.io), constructs an abstract syntax tree (AST) that represents the information contained in the MOD file. The `CodeGenerator` then uses the AST to produce a Python file from a JINJA template (https://jinja.palletsprojects.com). Users may also supply custom templates to generate specialized output files when required.

## Standardization of ion channel models

A part of the neuronal membrane can be modeled as an equivalent RC circuit. The membrane acts as a capacitor that can store charge, while ion channels provide resistive pathways for current flow. Voltage dynamics in this RC circuit are governed by the fundamental principle of current conservation (Kirchhoff's current law), which states that the capacitive current (left) must balance the sum of all ionic, synaptic, and external currents (right).

$$C\frac{dV}{dt} = \sum_{i=1}^{n} I_{ion,i}(t) + \sum_{j=1}^{m} I_{syn,j}(t) + I_{ext}(t) \tag{1}$$

where:

$V$ — the membrane potential in $mV$
$C$ — the membrane capacitance in $\mu F$
$I_{ion,i}$ — individual ionic currents in $nA$
$I_{syn,j}$ — individual synaptic currents in $nA$
$I_{ext}$ — external current in $nA$ (e.g. through a patch electrode)
$t$ — time in $ms$

The capacitive current term $C\frac{dV}{dt}$ is derived from the time derivative of the fundamental capacitance relationship $Q = CV$.

For a given ion channel, the current $I$ is determined by the channel's conductance state and the driving force:

$$I = \bar{g} \times p(x_1, \cdots, x_n) \times (V - E) \tag{2}$$

where:

$\bar{g}$ — the maximal channel conductance in $S$

$x_i$ — a state variable representing channel gating (unitless)

$p$ — the function defining the probability of the channel to be open (e.g. for an HH sodium channel $p(m, h) = m^3 h$)

$E$ — the equilibrium potential in $mV$

$(V - E)$ — the driving force in $mV$

The time derivative of a state variable $x$ is given by:

$$\frac{dx}{dt} = \frac{x^\infty - x}{\tau_x} \tag{3}$$

where:

$x^\infty$ — the steady-state value of $x$ (unitless)

$\tau_x$ — the time constant in $ms$

The voltage-dependent steady-state value $x^\infty$ is defined as:

$$x^\infty = \frac{1}{1 + \exp\left(-\dfrac{V - V_{half}}{\sigma}\right)} \tag{4}$$

The voltage-dependent time constant $\tau_x$ is given by:

$$\tau_x = \frac{1}{\dfrac{d\alpha}{dt} + \dfrac{d\beta}{dt}} + \tau_0 \tag{5}$$

where:

$$\frac{d\alpha}{dt} = K \times \exp\left(\frac{\delta \times (V - V_{half})}{\sigma}\right) \tag{6}$$

$$\frac{d\beta}{dt} = K \times \exp\left(\frac{-(1 - \delta) \times (V_{half} - V)}{\sigma}\right) \tag{7}$$

where:

$V$ — the membrane potential in $mV$

$V_{half}$ — the half-activation voltage in $mV$

$\sigma$ — the inverse slope in $mV$

$\delta$ — the skew parameter of the time constant curve (unitless)

$K$ — the maximum rate parameter in $ms^{-1}$

$\tau_0$ — the rate-limiting factor (minimum time constant) in $ms$

For each state variable of a channel, we fit the set of 5 parameters of the system of *Equations 4–7*, namely $V_{half}$, $\sigma$, $K$, $\delta$, and $\tau_0$, to the data in the form of activation (inactivation) curves derived from the original MOD files for membrane potentials in the range from –100 to 100 mV. The fitting process is implemented in the symfit Python library (https://symfit.readthedocs.io) to fit both curves for the steady state and the time constant simultaneously. The temperature is set to the original temperature before getting data to fit. We noticed that, while accurate for both curves, simultaneous fitting results in significant changes in the voltage and current dynamics. Therefore, we introduced a second additional fit for the steady state alone, sacrificing fitting accuracy for the time constant but preserving voltage and current dynamics. Finally, a new MOD file is created from a JINJA template and is immediately available to replace the original mechanism in the neuronal model.

## Morphology reduction

We extended the analytical impedance-based `neuron_reduce` approach proposed by *Amsalem et al., 2020* and integrated it into our GUI. The original `neuron_reduce` algorithm maps a dendritic subtree to a single cylinder with both ends sealed, preserving:

specific membrane resistivity, $R_m$ in $\Omega \times cm^2$
specific membrane capacitance, $C_m$ in $F/cm^2$
specific axial resistivity, $R_a$ in $\Omega \times cm$
transfer impedance from the electrotonically most distal dendritic tip to the soma, $\left|Z_{0,L}(\omega)\right|$
input resistance at the soma end (when disconnected from the soma), $\left|Z_{0,0}(\omega)\right|$

Equations 1 -11 in the original paper describe the unique cylindrical cable (with a specific diameter, $d$ and length, $L$, and the given membrane and axial properties) that preserves the values of $\left|Z_{0,L}(\omega)\right|$ and $\left|Z_{0,0}(\omega)\right|$. In the original implementation, the entire subtree of each stem dendrite (e.g. the entire apical subtree) is mapped to a single corresponding cylinder. We extended this approach to allow a user to select any section they want and map the inclusive subtree of this section (including the section itself) to a single cylinder. When the user selects the desired section using the GUI and clicks the button 'Reduce subtree' the inclusive subtree is disconnected from the cell and parameters for its equivalent cylinder are calculated. The exclusive subtree of the section is then removed, and the section's length and diameter are updated with the new calculated values before reconnecting it to its original parent. As in the original method, the reduced model is compartmentalized into segments (typically with a spatial resolution of 0.1 $\lambda$), and channel conductances are adjusted according to the mapping between the original and the reduced segments. In order to introduce a more general workflow, where synapses can be allocated on the already reduced model, we removed the step from the original algorithm that mapped synapses to the corresponding cylinder. To enable export of reduced models in *DendroTweaks*'s modular format and plain simulator code, we implemented a fitting procedure that approximates spatial parameter distributions using analytical functions (e.g., polynomials or step-like). Candidate models were fit to parameter values as a function of distance from the soma, and the optimal model was chosen by minimizing mean squared error, with model complexity as a secondary criterion. This approach provides a compact representation, independent of the original segmentation, requiring only a few coefficients to be stored to reproduce a distribution.

## Validation

We utilized a semi-automated approach for model validation. Users need to manually specify the simulation parameters for each specific validation protocol (see *Table 1*). Depending on the protocol, one or multiple simulation runs are performed, and the simulated voltage values are used for further calculations. Input resistance is calculated according to the formula: $R_{\text{in}} = \dfrac{V_{\text{offset}} - V_{\text{onset}}}{I_{\text{ext}}}$. The membrane time constant ($\tau$) is derived by fitting a double exponential equation to the decaying part of the voltage curve after the stimulus onset, taking the slowest component. Voltage attenuation is calculated for a user-specified set of segments by measuring the voltage response at different points along the dendrite. For each segment, the voltage attenuation is computed as the ratio of the voltage change at the segment ($\Delta V_{\text{seg}}$) to the voltage change at the stimulation site ($\Delta V_{\text{stim}}$). The distances from the soma to each segment are also recorded, and the attenuation is plotted against these distances. For detecting somatic action potentials and measuring their amplitude and half-width, we used the SciPy (*Virtanen et al., 2020*) Python library for peak detection in a signal. This involves identifying the peaks in the voltage trace and calculating the time difference between the points where the voltage is half of the peak amplitude. The somatic f-I curve is built by injecting a range of current amplitudes (e.g. 0–1 nA in steps of 0.1 nA) into the soma and recording the number of action potentials generated at each current level. The firing rate is plotted as a function of the injected current amplitude. Dendritic nonlinearities are derived by measuring the voltage response in a dendrite to increasing synaptic input weights of a single synapse. The unitary voltage response (EPSP) is determined, and the actual voltage responses for increasing synaptic weights are compared to the expected linear sum of unitary responses. The sag ratio is calculated according to the formula: Sag ratio $= \dfrac{a}{b}$, where $a = V_{\text{offset}} - V_{\text{min}}$ and $b = V_{\text{onset}} - V_{\text{min}}$. This ratio is derived from the voltage response to a hyperpolarizing current injection in the presence of HCN channels (h current), with $V_{\text{onset}}$ being the initial voltage before the injection, $V_{\text{min}}$ the minimum voltage reached, and $V_{\text{offset}}$ the voltage at the end of the injection.

## Acknowledgements

We thank members of the Poirazi lab and SmartNets ITN for their valuable feedback on the manuscript. This work was supported by NIH (1R01MH124867), the European Union, Horizon 2020 Programme

(H2020-FETOPEN-2018-2019-2020-01), NEUREKA (GA-863245), H2020 MSCA ITN Project SmartNets (GA-860949) and iNavigate (GA-873178), and the Stavros Niarchos Foundation (SNF) and the Hellenic Foundation for Research and Innovation (H.F.R.I.) under the 5th Call of "Science and Society" Action Always strive for excellence – Theodoros Papazoglou" grant DendroLeap (Project Number: 28056).

## Additional information

### Competing interests

Panayiota Poirazi: Senior editor, eLife. The other authors declare that no competing interests exist.

### Funding

| Funder | Grant reference number | Author |
| --- | --- | --- |
| National Institutes of Health | 1R01MH124867 | Panayiota Poirazi |
| Horizon 2020 Framework Programme | GA-863245 | Panayiota Poirazi |
| Horizon 2020 Framework Programme | GA-860949 | Panayiota Poirazi |
| Horizon 2020 Framework Programme | GA-873178 | Panayiota Poirazi |
| Hellenic Foundation for Research and Innovation - Stavros Niarchos Foundation | GA-28056 | Panayiota Poirazi |

The funders had no role in study design, data collection and interpretation, or the decision to submit the work for publication.

### Author contributions

Roman Makarov, Software, Formal analysis, Validation, Investigation, Visualization, Methodology, Writing - original draft; Spyridon Chavlis, Formal analysis, Supervision, Methodology, Writing – review and editing; Panayiota Poirazi, Conceptualization, Supervision, Funding acquisition, Methodology, Writing – review and editing

### Author ORCIDs

Roman Makarov ⓘ http://orcid.org/0000-0002-4174-3826
Spyridon Chavlis ⓘ http://orcid.org/0000-0002-1046-1201
Panayiota Poirazi ⓘ https://orcid.org/0000-0001-6152-595X

Reviewer #1 (Public review): https://doi.org/10.7554/eLife.103324.3.sa1
Reviewer #2 (Public review): https://doi.org/10.7554/eLife.103324.3.sa2
Author response https://doi.org/10.7554/eLife.103324.3.sa3

## Additional files

### Supplementary files

MDAR checklist

### Data availability

The current manuscript is a computational study, so no data have been generated for this manuscript. The source code is openly available on GitHub for both the standalone Python package https://github.com/Poirazi-Lab/DendroTweaks (*Makarov et al., 2025a*) and the web application https://github.com/Poirazi-Lab/DendroTweaksApp (*Makarov et al., 2025b*). Detailed documentation and tutorials are available through ReadTheDocs (https://dendrotweaks.readthedocs.io/en/latest).

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
