## [Editor Report · eLife Assessment]

Computational simulation of neuron function depends on a collection of morphological properties and ion channel biophysics. This manuscript introduces DendroTweaks, a **valuable** web application and Python library that eases interactive exploration, development, and validation of single-neuron models in an easily installable and well-documented package. The authors provide a **convincing** demonstration that their software aids with building intuition and rapid prototyping of biophysical models of neurons, which improves the accessibility of dendritic simulation.

---

## [Referee Report · Reviewer #1 (Public review)]

Summary:

Dendrotweaks provides to its users a solid tool to implement, visualize, tune, validate, understand, and reduce single-neuron models that incorporate complex dendritic arbors with differential distribution of biophysical mechanisms. The visualization of dendritic segments and biophysical mechanisms therein provide users an intuitive way to understand and appreciate dendritic physiology.

---

## [Referee Report · Reviewer #2 (Public review)]

The paper by Makarov et al. describes the software tool called DendroTweaks, intended for examination of multi-compartmental biophysically detailed neuron models. It offers extensive capabilities for working with very complex distributed biophysical neuronal models and should be a useful addition to the growing ecosystem of tools for neuronal modeling.

Strengths

• This Python-based tool allows for visualization of a neuronal model's compartments.

• The tool works with morphology reconstructions in the widely used .swc and .asc formats.

• It can support many neuronal models using the NMODL language, which is widely used for neuronal modeling.

• It permits one to plot the properties of linear and non-linear conductances in every compartment of a neuronal model, facilitating examination of model's details.

• DendroTweaks supports manipulation of the model parameters and morphological details, which is important for exploration of the relations of the model composition and parameters with its electrophysiological activity.

• The paper is very well written - everything is clear, and the capabilities of the tool are described and illustrated with great attention to details.

Weaknesses

• Not a really big weakness, but it would be really helpful if the authors showed how the performance of their tool scales. This can be done for an increasing number of compartments - how long does it take to carry out typical procedures in DendroTweaks, on a given hardware, for a cell model with 100 compartments, 200, 300, and so on? This information will be quite useful to understand the applicability of the software.

Let me also add here a few suggestions (not weaknesses, but something that can be useful, and if the authors can easily add some of these for publication, that would strongly increase the value of the paper).

• It would be very helpful to add functionality to read major formats in the field, such as NeuroML and SONATA.

• Visualization is available as a static 2D projection of the cell's morphology. It would be nice to implement 3D interactive visualization.

• It is nice that DendroTweaks can modify the models, such as revising the radii of the morphological segments or ionic conductances. It would be really useful then to have the functionality for writing the resulting models into files for subsequent reuse.

• If I didn't miss something, it seems that DendroTweaks supports allocation of groups of synapses, where all synapses in a group receive the same type of Poisson spike train. It would be very useful to provide more flexibility. One option is to leverage the SONATA format, which has ample functionality for specifying such diverse inputs.

• "Each session can be saved as a .json file and reuploaded when needed" - do these files contain the whole history of the session or the exact snapshot of what is visualized when the file is saved? If the latter, which variables are saved, and which are not? Please clarify.

Comments on revisions:

In this revised version of the paper, the authors addressed all my comments. While many of the suggestions were addressed by textual changes in the manuscript or an explanation in the response to the reviewers (rather than adding substantial new functionality to the tool), DendroTweaks in its current updated state does represent an advanced and useful tool. Further extensions can be added as the development of the tool continues, in interaction with the community.

---

## [Author Response]

The following is the authors’ response to the original reviews.

**Reviewer #1 (Public review):**
Summary:Dendrotweaks provides its users with a solid tool to implement, visualize, tune, validate, understand, and reduce single-neuron models that incorporate complex dendritic arbors with differential distribution of biophysical mechanisms. The visualization of dendritic segments and biophysical mechanisms therein provide users with an intuitive way to understand and appreciate dendritic physiology.Strengths:(1) The visualization tools are simplified, elegant, and intuitive.(2) The ability to build single-neuron models using simple and intuitive interfaces.(3) The ability to validate models with different measurements.(4) The ability to systematically and progressively reduce morphologically-realistic neuronal models.Weaknesses:(1) Inability to account for neuron-to-neuron variability in structural, biophysical, and physiological properties in the model-building and validation processes.

We agree with the reviewer that it is important to account for neuron-to-neuron variability. The core approach of DendroTweaks, and its strongest aspect, is the interactive exploration of how morpho-electric parameters affect neuronal activity. In light of this, variability can be achieved through the interactive updating of the model parameters with widgets. In a sense, by adjusting a widget (e.g., channel distribution or kinetics), a user ends up with a new instance of a cell in the parameter space and receives almost real-time feedback on how this change affected neuronal activity. This approach is much simpler than implementing complex optimization protocols for different parameter sets, which would detract from the interactivity aspect of the GUI. In its revised version, DendroTweaks also accounts for neuron-to-neuron morphological variability, as channel distributions are now based on morphological domains (rather than the previous segment-specific approach). This makes it possible to apply the same biophysical configuration across various morphologies. Overall, both biophysical and morphological variability can be explored within DendroTweaks.

(2) Inability to account for the many-to-many mapping between ion channels and physiological outcomes. Reliance on hand-tuning provides a single biased model that does not respect pronounced neuron-to-neuron variability observed in electrophysiological measurements.

We acknowledge the challenge of accounting for degeneracy in the relation between ion channels and physiological outcomes and the importance of capturing neuron-to-neuron variability. One possible way to address this, as we mention in the Discussion, is to integrate automated parameter optimization algorithms alongside the existing interactive hand-tuning with widgets. In its revised version, DendroTweaks can integrate with Jaxley (Deistler et al., 2024) in addition to NEURON. The models created in DendroTweaks can now be run with Jaxley (although not all types of models, see the limitations in the Discussion), and their parameters can be optimized via automated and fast gradient-based parameter optimization, including optimization of heterogeneous channel distributions. In particular, a key advantage of integrating Jaxley with DendroTweaks was its NMODL-to-Python converter, which significantly reduced the need to manually re-implement existing ion channel models for Jaxley (see here: https://dendrotweaks.readthedocs.io/en/latest/tutorials/convert_to_jaxley.html).

(1) Michael Deistler, Kyra L. Kadhim, Matthijs Pals, Jonas Beck, Ziwei Huang, Manuel Gloeckler, Janne K. Lappalainen, Cornelius Schröder, Philipp Berens, Pedro J. Gonçalves, Jakob H. Macke Differentiable simulation enables large-scale training of detailed biophysical models of neural dynamics bioRxiv 2024.08.21.608979; doi:https://doi.org/10.1101/2024.08.21.608979

Lack of a demonstration on how to connect reduced models into a network within the toolbox.

Building a network of reduced models is an exciting direction, yet beyond the scope of this manuscript, whose primary goal is to introduce DendroTweaks and highlight its capabilities. DendroTweaks is designed for single-cell modeling, aiming to cover its various aspects in great detail. Of course, we expect refined single-cell models, both detailed and simplified, to be further integrated into networks. But this does not need to occur within DendroTweaks. We believe this network-building step is best handled by dedicated network simulation platforms. To facilitate the network-building process, we extended the exporting capabilities of DendroTweaks. To enable the export of reduced models in DendroTweaks’s modular format, as well as in plain simulator code, we implemented a method to fit the resulting parameter distributions to analytical functions (e.g., polynomials). This approach provided a compact representation, requiring a few coefficients to be stored in order to reproduce a distribution, independently of the original segmentation. The reduced morphologies can be exported as SWC files, standardized ion channel models as MOD files, and channel distributions as JSON files. Moreover, plain NEURON code (Python) to instantiate a cell class can be automatically generated for any model, including the reduced ones. Finally, to demonstrate how these exported models can be integrated into larger simulations, we implemented a "toy" network model in a Jupyter notebook included as an example in the GitHub repository. We believe that these changes greatly facilitate the integration of DendroTweaks-produced models into networks while also allowing users to run these networks on their favorite platforms.

(4) Lack of a set of tutorials, which is common across many "Tools and Resources" papers, that would be helpful in users getting acquainted with the toolbox.

This is an important point that we believe has been addressed fully in the revised version of the tool and manuscript. As previously mentioned, the lack of documentation was due to the software's early stage. We have now added comprehensive documentation, which is available at https://dendrotweaks.readthedocs.io. This extensive material includes API references, 12 tutorials, 4 interactive Jupyter notebooks, and a series of video tutorials, and it is regularly updated with new content. Moreover, the toolbox's GUI with example models is available through our online platform at https://dendrotweaks.dendrites.gr.

**Reviewer #2 (Public review):**
The paper by Makarov et al. describes the software tool called DendroTweaks, intended for the examination of multi-compartmental biophysically detailed neuron models. It offers extensive capabilities for working with very complex distributed biophysical neuronal models and should be a useful addition to the growing ecosystem of tools for neuronal modeling.Strengths(1) This Python-based tool allows for visualization of a neuronal model's compartments.(2) The tool works with morphology reconstructions in the widely used .swc and .asc formats.(3) It can support many neuronal models using the NMODL language, which is widely used for neuronal modeling.(4) It permits one to plot the properties of linear and non-linear conductances in every compartment of a neuronal model, facilitating examination of the model's details.(5) DendroTweaks supports manipulation of the model parameters and morphological details, which is important for the exploration of the relations of the model composition and parameters with its electrophysiological activity.(6) The paper is very well written - everything is clear, and the capabilities of the tool are described and illustrated with great attention to detail.Weaknesses(1) Not a really big weakness, but it would be really helpful if the authors showed how the performance of their tool scales. This can be done for an increasing number of compartments - how long does it take to carry out typical procedures in DendroTweaks, on a given hardware, for a cell model with 100 compartments, 200, 300, and so on? This information will be quite useful to understand the applicability of the software.

DendroTweaks functions as a layer on top of a simulator. As a result, its performance scales in the same way as for a given simulator. The GUI currently displays the time taken to run a simulation (e.g., in NEURON) at the bottom of the Simulation tab in the left menu. While Bokeh-related processing and rendering also consume time, this is not as straightforward to measure. It is worth noting, however, that this time is short and approximately equivalent to rendering the corresponding plots elsewhere (e.g., in a Jupyter notebook), and thus adds negligible overhead to the total simulation time.

(2) Let me also add here a few suggestions (not weaknesses, but something that can be useful, and if the authors can easily add some of these for publication, that would strongly increase the value of the paper).(3) It would be very helpful to add functionality to read major formats in the field, such as NeuroML and SONATA.

We agree with the reviewer that support for major formats will substantially improve the toolbox, ensuring the reproducibility and reusability of the models. While integration with these formats has not been fully implemented, we have taken several steps to ensure elegant and reproducible model representation. Specifically, we have increased the modularity of model components and developed a custom compact data format tailored to single-cell modeling needs. We used a JSON representation inspired by the Allen Cell Types Database schema, modified to account for non-constant distributions of the model parameters. We have transitioned from a representation of parameter distributions dependent on specific segmentation graphs and sections to a more generalized domain-based distribution approach. In this revised methodology, segment groups are no longer explicitly defined by segment identifiers, but rather by specification of anatomical domains and conditional expressions (e.g., “select all segments in the apical domain with the maximum diameter < 0.8 µm”). Additionally, we have implemented the export of experimental protocols into CSV and JSON files, where the JSON files contain information about the stimuli (e.g., synaptic conductance, time constants), and the CSV files store locations of recording sites and stimuli. These features contribute toward a higher-level, structured representation of models, which we view as an important step toward eventual compatibility with standard formats such as NeuroML and SONATA. We have also initiated a two-way integration between DendroTweaks and SONATA. We developed a converter from DendroTweaks to SONATA that automatically generates SONATA files to reproduce models created in DendroTweaks. Additionally, support for the DendroTweaks JSON representation of biophysical properties will be added to the SONATA data format ecosystem, enabling models with complex dendritic distributions of channels. This integration is still in progress and will be included in the next version of DendroTweaks. While full integration with these formats is a goal for future releases, we believe the current enhancements to modularity and exportability represent a significant step forward, providing immediate value to the community.

(4) Visualization is available as a static 2D projection of the cell's morphology. It would be nice to implement 3D interactive visualization.

We offer an option to rotate a cell around the Y axis using a slider under the plot. This is a workaround, as implementing a true 3D visualization in Bokeh would require custom Bokeh elements, along with external JavaScript libraries. It's worth noting that there are already specialized tools available for 3D morphology visualization. In light of this, while a 3D approach is technically feasible, we advocate for a different method. The core idea of DendroTweaks’ morphology exploration is that each section is “clickable”, allowing its geometric properties to be examined in a 2D "Section" view. Furthermore, we believe the "Graph" view presents the overall cell topology and distribution of channels and synapses more clearly.

(5) It is nice that DendroTweaks can modify the models, such as revising the radii of the morphological segments or ionic conductances. It would be really useful then to have the functionality for writing the resulting models into files for subsequent reuse.

This functionality is fully available in local installations. Users can export JSON files with channel distributions and SWC files after morphology reduction through the GUI. Please note that for resource management purposes, file import/export is disabled on the public online demo. However, it can be enabled upon local installation by modifying the configuration file (app/default_config.json). In addition, it is now possible to generate plain NEURON (Python) code to reproduce a given model outside the toolbox (e.g., for network simulations). Moreover, it is now possible to export the simulation protocols as CSV files for locations of stimuli and recordings and JSON files for stimuli parameters.

(6) If I didn't miss something, it seems that DendroTweaks supports the allocation of groups of synapses, where all synapses in a group receive the same type of Poisson spike train. It would be very useful to provide more flexibility. One option is to leverage the SONATA format, which has ample functionality for specifying such diverse inputs.

Currently, each population of “virtual” neurons that form synapses on the detailed cell shares the same set of parameters for both biophysical properties of synapses (e.g., reversal potential, time constants) and presynaptic "population" activity (e.g., rate, onset). The parameter that controls an incoming Poisson spike train is the rate, which is indeed shared across all synapses in a population. Unfortunately, the current implementation lacks the capability to simulate complex synaptic inputs with heterogeneous parameters across individual synapses or those following non-uniform statistical distributions (the present implementation is limited to random uniform distributions). We have added this information in the Discussion (3. Discussion - 3.2 Limitations and future directions - ¶.5) to make users aware of the limitations. As it requires a substantial amount of additional work, we plan to address such limitations in future versions of the toolbox.

(7) "Each session can be saved as a .json file and reuploaded when needed" - do these files contain the whole history of the session or the exact snapshot of what is visualized when the file is saved? If the latter, which variables are saved, and which are not? Please clarify.

In the previous implementation, these files captured the exact snapshot of the model's latest state. In the new version, we adopted a modular approach where the biophysical configuration (e.g., channel distributions) and stimulation protocols are exported to separate files. This allows the user to easily load and switch the stimulation protocols for a given model. In addition, the distribution of parameters (e.g., channel conductances) is now based on the morphological domains and is agnostic of the exact morphology (i.e., sections and segments), which allows the same JSON files with biophysical configurations to be reused across multiple similar morphologies. This also allows for easy file exchange between the GUI and the standalone version.

**Joint recommendations to Authors:**
The reviewers agreed that the paper is well written and that DendroTweaks offers a useful collection of tools to explore models of single-cell biophysics. However, the tooling as provided with this submission has critical limitations in the capabilities, accessibility, and documentation that significantly limit the utility of DendroTweaks. While we recognize that it is under active development and features may have changed already, we can only evaluate the code and documentation available to us here.

We thank the reviewers for their positive evaluation of the manuscript and express our sincere appreciation for their feedback. We acknowledge the limitations they have pointed out and have addressed most of these concerns in our revised version.

In particular, we would emphasize:(1) While the features may be rich, the documentation for either a user of the graphical interface or the library is extremely sparse. A collection of specific tutorials walking a GUI user through simple and complex model examples would be vital for genuine uptake. As one category of the intended user is likely to be new to computational modeling, it would be particularly good if this documentation could also highlight known issues that can arise from the naive use of computational techniques. Similarly, the library aspect needs to be documented in a more standard manner, with docstrings, an API function list, and more didactic tutorials for standard use cases.

DendroTweaks now features comprehensive documentation. The standalone Python library code is well-documented with thorough docstrings. The overall code modularity and readability have improved. The documentation is created using the widely adopted Sphinx generator, making it accessible for external contributors, and it is available via ReadTheDocs https://dendrotweaks.readthedocs.io/en/latest/index.html. The documentation provides a comprehensive set of tutorials (6 basic, 6 advanced) covering all key concepts and workflows offered by the toolbox. Interactive Jupyter notebooks are included in the documentation, along with the quick start guide. All example models also have corresponding notebooks that allow users to build the model from scratch.

The toolbox has its own online platform, where a quick-start guide for the GUI is available https://dendrotweaks.dendrites.gr/guide.html. We have created video tutorials for the GUI covering the basic use cases. Additionally, we have added tips and instructions alongside widgets in the GUI, as well as a status panel that displays application status, warnings, and other information. Finally, we plan to familiarize the community with the toolbox by organizing online and in-person tutorials, as the one recently held at the CNS*2025 conference (https://cns2025florence.sched.com/event/25kVa/building-intuitive-and-efficient-biophysicalmodels-with-jaxley-and-dendrotweaks). Moreover, the toolbox was already successfully used for training young researchers during the Taiwan NeuroAI 2025 Summer School, founded by Ching-Lung Hsu. The feedback was very positive.

(2) The paper describes both a GUI web app and a Python library. However, the code currently mixes these two in a way that largely makes sense for the web app but makes it very difficult to use the library aspect. Refactoring the code to separate apps and libraries would be important for anyone to use the library as well as allowing others to host their own DendroTweak servers. Please see the notes from the reviewing editor below for more details.

The code in the previous `app/model` folder, responsible for the core functionality of the toolbox, has been extensively refactored and extended, and separated into a standalone library. The library is included in the Python package index (PyPI, https://pypi.org/project/dendrotweaks).

Notes from the Reviewing Editor Comments (Recommendations for the authors):(1) While one could import morphologies and use a collection of ion channel models, details of synapse groups and stimulation approaches appeared to be only configurable manually in the GUI. The ability to save and load full neuron and simulation states would be extremely useful for reproducibility and sharing data with collaborators or as an interactive data product with a publication. There is a line in the text about saving states as json files (also mentioned by Reviewer #2), but I could see no such feature in the version currently online.

We decided to reserve the online version for demonstration and educational purposes, with more example models being added over time. However, this functionality is available upon local installation of the app (and after specifying it in the ‘default_config.json’ in the root directory of the app). We’ve adopted a modular model representation to store separately morphology, channel models, biophysical parameters, and stimulation protocols.

(2) Relatedly, GUI exploration of complex data is often a precursor to a more automated simulation run. An easy mechanism to go from a user configuration to scripting would be useful to allow the early strength of GUIs to feed into the power of large-scale scripting.

Any model could be easily exported to a modular DendroTweaks representation and later imported either in the GUI or in the standalone version programmatically. This ensures a seamless transition between the two use cases.

(3) While the paper discusses DendroTweaks as both a GUI and a python library, the zip file of code in the submission is not in good form as a library. Back-end library code is intermingled with front-end web app code, which limits the ability to install the library from a standard python interface like PyPI. API documentation is also lacking. Functions tend to not have docstrings, and the few that do, do not follow typical patterns describing parameters and types.

As stated above, all these issues have been resolved in the new version of the toolbox. The library code is now housed in a separate repository https://github.com/Poirazi-Lab/DendroTweaks and included in PyPI https://pypi.org/project/dendrotweaks. The classes and public methods follow Numpy-style docstrings, and the API reference is available in the documentation: https://dendrotweaks.readthedocs.io/en/latest/genindex.html.

(4) Library installation is very difficult. The requirements are currently a lockfile, fully specifying exact versions of all dependencies. This is exactly correct for web app deployment to maintain consistency, but is not feasible in the context of libraries where you want to have minimal impact on a user's environment. Refactoring the library from the web app is critical for making DendroTweaks usable in both forms described in the paper.The lockfile makes installation more or less impossible on computer setups other than that of the author. Needless to say, this is not acceptable for a tool, and I would encourage the authors to ask other people to attempt to install their code as they describe in the text. For example, attempting to create a conda environment from the environment.yml file on an M1 MacBook Pro failed because it could not find several requirements. I was able to get it to install within a Linux docker image with the x86 platform specified, but this is not generally viable. To make this be the tool it is described as in text, this must be resolved. A common pattern that would work well here is to have a requirements lockfile and Docker image for the web app that imports a separate, more minimally restrictive library package with that could be hosted on PyPI or, less conveniently, through conda-forge.

The installation of the standalone library is now straightforward via pip install dendrotweaks.On the Windows platform, however, manual installation of NEURON is required as described in the official NEURON documentation https://nrn.readthedocs.io/en/8.2.6/install/install_instructions.html#windows.

(5) As an aside, to improve potential uptake, the authors might consider an MIT-style license rather than the GNU Public License unless they feel strongly about the GPL. Many organizations are hesitant to build on GPL software because of the wide-ranging demands it places on software derived from or using GPL code.

We thank the editor for this suggestion. We are considering changing the licence to MPL 2.0. It will maintain copyleft restrictions only on the package files while allowing end-users to freely choose their own license for any derived work, including the models, generated data files, and code that simply imports and uses our package.

**Reviewer #1 (Recommendations for the authors):**
(1) Abstract: Neurons rely on the interplay between dendritic morphology and ion channels to transform synaptic inputs into a sequence of somatic spikes. Technically, this would have to be morphology, ion channels, pumps, transporters, exchangers, buffers, calcium stores, and other molecules. For instance, if the calcium buffer concentration is large, then there would be less free calcium for activating the calcium-activated potassium channels. If there are different chloride co-transporters - NKCC vs. KCC - expressed in the neuron or different parts of the neuron, that would alter the chloride reversal for all the voltage- or ligand-gated chloride channels in the neuron. So, while morphology and ion channels are two important parts of the transformation, it would be incorrect to ignore the other components that contribute to the transformation. The statement might be revised to make these two components as two critical components.

The phrase “Two critical components” was added as it was suggested by the reviewer.

(2) Section 2.1 - The overall GUI looks intuitive and simple.(3) Section 2.2(a) The Graph view of morphology, especially accounting for the specific d_lambda is useful.(b) "Note that while microgeometry might not significantly affect the simulation at a low spatial resolution (small number of segments) due to averaging, it can introduce unexpected cell behavior at a higher level of spatial discretization."It might be good to warn the users that the compartmentalization and error analyses are with reference to the electrical lambda. If users have to account for calcium microdomains, these analyses wouldn't hold given the 2 orders of magnitude differences between the electrical and the calcium lambdas (e.g., Zador and Koch, J Neuroscience, 1994). Please sensitize users that the impact of active dendrites in regulating calcium microdomains and signaling is critical when it comes to plasticity models in morphologically realistic structures.

We thank the reviewer for this important point. We have clarified in the text that our spatial discretization specifically refers to the electrical length constant. We acknowledge that electrical and chemical processes operate on fundamentally different spatial and temporal scales, which requires special consideration when modeling phenomena like synaptic plasticity. We have sensitized users about this distinction. However, we do not address such examples in the manuscript, thus leaving the detailed discussion of non-electrical compartmentalization beyond the scope of this work.

(c) I am not very sure if the "smooth" tool for diameters that is illustrated is useful. Users shouldn't consider real variability in morphology as artifacts of reconstruction. As mentioned above, while this might not be an issue with electrical compartmentalization, calcium compartmentalization will severely be affected by small changes in morphology. Any model that incorporates calcium-gated channels should appropriately compartmentalize calcium. Without this, the spread of activation of calcium-dependent conductances would be an overestimate. Even small changes in cellular shape and curvature can have large impacts when it comes to signaling in terms of protein aggregation and clustering.

Although this functionality is still available in the toolbox, we have removed the emphasis from it in the manuscript. Nevertheless, for the purpose of addressing the reviewer’s comment, we provide an example when this “smoothening” might be needed:please see Figure S1 from Tasciotti et al. 2025.

(2) Simone Tasciotti, Daniel Maxim Iascone, Spyridon Chavlis, Luke Hammond, Yardena Katz, Attila Losonczy, Franck Polleux, Panayiota Poirazi. From Morphology to Computation: How Synaptic Organization Shapes Place Fields in CA1 Pyramidal Neurons bioRxiv 2025.05.30.657022; doi: https://doi.org/10.1101/2025.05.30.657022

(4) Section 2.3(a) The graphical representation of channel gating kinetics is very useful.(b) Please warn the users that experimental measurements of channel gating kinetics are extremely variable. Taking the average of the sigmoids or the activation/deactivation/inactivation kinetics provides an illusion that each channel subtype in a given cell type has fixed values of V_1/2, k, delta, and tau, but it is really a range obtained from several experiments. The heterogeneity is real and reflects cell-to-cell variability in channel gating kinetics, not experimental artifacts. Please sensitize the readers that there is not a single value for these channel parameters.

This is a fair comment, and it refers to a general problem in neuronal modeling. In DendroTweaks, we follow the approach widely used in the community that indeed doesn't account for heterogeneity. We added a paragraph in the revised manuscript's Discussion (3. Discussion - 3.3 Limitations and future directions - ¶.3) to address this issue.

(5) Section 2.4(a) Same as above: Please sensitize users that the gradients in channel conductances are measured as an average of measurements from several different cells. This gradient need not be present in each neuron, as there could be variability in location-dependent measurements across cells. The average following a sigmoid doesn't necessarily mean that each neuron will have the channel distributed with that specific sigmoid (or even a sigmoid!) with the specific parametric values that the average reported. This is extremely important because there is an illusion that the gradient is fixed across cells and follows a fixed functional form.

We added this information to our Discussion in the same paragraph mentioned above.

(b) Please provide an example where the half-maximal voltage of a channel varies as a function of distance (such as Poolos et al., Nature Neuroscience, 2002 or Migliore et al., 1999; Colbert and Johnston, 1997). This might require a step-like function in some scenarios. An illustration would be appropriate because people tend to assume that channel gating kinetics are similar throughout the dendrite. Again, please mention that these shifts are gleaned from the average and don't really imply that each neuron must have that specific gradient, given neuron-to-neuron variability in these measurements.

We thank the reviewer for the provided literature, which we now cite when describing parameter distributions (2. Results - 2.4 Distributing ion channels - ¶.1). Please note that DendroTweaks' programming interface and data format natively support non-linear distribution of kinetic parameters alongside the channel conductances. As for the step-like function, users can either directly apply the built-in step-like distribution function or create it by combining two constant distributions.

(6) Section 2.5(a) It might be useful to provide a mechanism for implementing the normalization of unitary conductances at the cell body, (as in Magee and Cook, 2000; Andrasfalvy et al., J Neuroscience, 2001). Specifically, users should be able to compute AMPAR conductance values at each segment which would provide a somatic EPSP value of 0.2 mV.

This functionality is indeed useful and will be added in future releases. Currently, it has been mentioned in the list of known limitations when working with synaptic inputs (3. Discussion - 3.3 Limitations and future directions - ¶.5).

(b) Users could be sensitized about differences in decay time constants of GABA_A receptors that are associated with parvalbamin vs. somatostatin neurons. As these have been linked to slow and fast gamma oscillations and different somatodendritic locations along different cell types, this might be useful (e.g., 10.1016/j.neuron.2017.11.033;10.1523/jneurosci.0261-20.2020; 10.7554/eLife.95562.1; 10.3389/fncel.2023.1146278).

We thank the reviewer for highlighting this important biological detail. DendroTweaks enables users to define model parameters specific to their cell type of interest. For practical reasons, we leave the selection of biologically relevant parameters to the users. However, we will consider adding an explicit example in our tutorials to showcase the toolbox's flexibility in this regard.

(7) Section 2.6While reducing the morphological complexity has its advantages, users of this tool should be sensitized in this section about how the reduction does not capture all the complexity of the dendritic computation. For instance, the segregation/amplification properties of Polsky et al., 2004, Larkum et al., 2009 would not be captured by a fully reduced model. An example across different levels of reductions, implementing simulations in Figure 7F (but for synapses on the same vs. different branches), would be ideal. Demonstrate segregation/amplification in the full model for the same set of synapses - coming on the same branch/different branch (linear integration of synapses on different branches and nonlinear integration of synapses on the same branch). Then, show that with different levels of reduction, this segregation/amplification vanishes in the reduced model. In addition, while impedance-based approaches account for account for electrical computation, calcium-based computation is not something that is accountable with reduced models, given the small lambda_calcium values. Given the importance of calcium-activated conductances in electrical behaviour, this becomes extremely important to account for and sensitize users to. The lack of such sensitization results in presumptuous reductions that assume that all dendritic computation is accounted for by reduced models!

We agree with the reviewer that reduction leads to a loss in the complexity of dendritic computation. This has been stated in both the original algorithm paper (Amsalem et al., 2020) and in our manuscript (e.g., 3. Discussion - 3.2 Comparison to existing modeling software - ¶.6). In fact, to address this problem, we extended the functionality of neuron_reduce to allow for multiple levels of morphology reduction. Our motivation for integrating morphology reduction in the toolbox was to leverage the exploratory power of DendroTweaks to assess how different degrees of reduction alter cell integrative properties, determining which computations are preserved, which are lost, and at what specific reduction level these changes occur. Nevertheless, to address this comment, we've made it more explicit in the Discussion that reduction inevitably alters integrative properties and, at a certain level, leads to loss of dendritic computations.

(8) Section 2.7(a) The validation process has two implicit assumptions:(i) There is only one value of physiological measurements that neurons and dendrites are endowed with. The heterogeneity in these measurements even within the same cell type is ignored. The users should be allowed to validate each measurement over a range rather than a single value. Users should be sensitized about the heterogeneity of physiological measurements.(ii) The validation process is largely akin to hand-tuning models where a one-to-one mapping of channels to measurements is assumed. For instance, input resistance can be altered by passive properties, by Ih, and by any channel that is active under resting conditions. Firing rate and patterns can be changed by pretty much every single ion channel that expresses along the somatodendritic axis.An updated validation process that respects physiological heterogeneities in measurements and accounts for global dependencies would be more appropriate. Please update these to account for heterogeneities and many-to-many mappings between channels and measurements. An ideal implementation would be to incorporate randomized search procedures (across channel parameters spanning neuron-to-neuron variability in channel conductances/gating properties) to find a population of models that satisfy all physiological constraints (including neuron-to-neuron variability in each physiological measurement), rather than reliance on procedures that are akin to hand-tuning models. Such population-based approaches are now common across morphologically-realistic models for different cell types (e.g., Rathour and Narayanan, PNAS, 2014; Basak and Narayanan, J Physiology, 2018; Migliore et al., PLoS Computational Biology, 2018; Basak and Narayanan, Brain Structure and Function, 2020; Roy and Narayanan, Neural Networks, 2021; Roy and Narayanan, J Physiology, 2023; Arnaudon et al., iScience, 2023; Reva et al., Patterns, 2023; Kumari and Narayanan, J Neurophysiology, 2024) and do away with the biases introduced by hand-tuning as well as the assumption of one-to-one mapping between channels and measurements.

We appreciate the reviewer’s comment and the suggested alternatives to our validation process. We have extended the discussion on these alternative approaches (3. Discussion - 2. Comparison to existing modeling software - ¶.5). However, it is important to note that neither one-value nor one-to-one mapping assumption is imposed in our approach. It is true that validation is performed on a given model instance with fixed single-value parameters. However, users can discover heterogeneity and degeneracy in their models via interactive exploration. In the GUI, a given parameter can be changed, and the influence of this change on model output can be observed in real time. Validation can be run after each change to see whether the model output still falls within a biologically plausible regime or not. This is, of course, time-consuming and less efficient than any automated parameter optimization.

However, and importantly, this is the niche of DendroTweaks. The approach we provide here can indeed be referred to as model hand-tuning. This is intentional: we aim to complement black-box optimization by exposing the relationship between parameters and model outputs. DendroTweaks is not aimed at automated parameter optimization and is not meant to provide the user with parameter ranges automatically. The built-in validation in DendroTweaks is intended as a lightweight, fast feedback tool to guide manual tuning of dendritic model parameters so as to enhance intuitive understanding and assess the plausibility of outputs, not as a substitute for comprehensive model validation or optimization. The latter can be done using existing frameworks, designed for this purpose, as mentioned by the reviewer.

(b) Users could be asked to wait for RMP to reach steady state. For instance, in some of the traces in Figure 7, the current injection is provided before RMP reaches steady-state. In the presence of slow channels (HCN or calcium-activated channels), the RMP can take a while to settle down. Users might be sensitized about this. This would also bring to attention the ability of several resting channels in modulating RMP, and the need to wait for steady-state before measurements are made.

We agree with the observation and updated the validation process accordingly. We have added functionality for simulation stabilization, allowing users to pre-run a simulation before the main simulation time. For example, model.run(duration=1000, prerun_time=300) could be used to stabilize the model for a period of 300 ms before running the main simulation for 1 s.

(c) Strictly speaking, it is incorrect to obtain membrane time constant by fitting a single exponential to the initial part of the sag response (Figure 7A). This may be confirmed in the model by setting HCN to zero (strictly all active channel conductances to zero), obtaining the voltage-response to a pulse current, fitting a double exponential (as Rall showed, for a finite cable or for a real neuron, a single exponential would yield incorrect values for the tau) to the voltage response, and mapping membrane time constant to the slower of the two time-constants (in the double exponential fit). This value will be very different from what is obtained in Figure 7A. Please correct this, with references to Rall's original papers and to electrophysiological papers that use this process to assess membrane properties of neurons and their dendrites (e.g., Stuart and Spruston, J Neurosci, 1998; Golding and Spruston, J Physiology, 2005).

We updated the algorithm for calculating the membrane time constant based on the reviewer's suggestions and added the suggested references. The time constant is now obtained in a model with blocked HCN channels (setting maximal conductance to 0) via a double exponential fit, taking the slowest component.

(9) Section 3(a) May be good to emphasize the many-to-many mapping between ion channels and neuronal functions here in detail, and on how to explore this within the Dendrotweaks framework.

We have added a paragraph in the Discussion that addresses both the problems of heterogeneity and degeneracy in biological neurons and neuronal models (3. Discussion - 3.3 Limitations and future directions - ¶.3)

(b) May be good to have a specific section either here or in results about how the different reduced models can actually be incorporated towards building a network.

As mentioned earlier, building a network of reduced models is a promising new direction. However, it is beyond the scope of this manuscript, whose primary goal is to introduce DendroTweaks and highlight its capabilities. DendroTweaks is designed for single-cell modeling and provides export capabilities that allow integrating it into broader workflows, including network modeling. We have added a paragraph in the manuscript (3. Discussion - 3.1 Conceptual and implementational accessibility - ¶.2) that addresses how DendroTweaks could be used alongside other software, in particular for scaling up single-cell models to the network level.

(10) Section 4(a) Section 4.3: In the second sentence (line 568), the "first Kirchhoff's law" within parentheses immediately after Q=CV gives an illusion that Q=CV is the first Kirchhoff's law! Please state that this is with reference to the algebraic sum of currents at a node.

We have corrected the equations and apologize for this oversight.

(b) Table 1: In the presence of active ion channels, input resistance, membrane time constant, and voltage attenuation are not passive properties. Input resistance is affected by any active channel that is active at rest (HCN, Kir, A-type K+ through the window current, etc). The same holds for membrane time constant and voltage attenuation as well. This could be made clear by stating if these measurements are obtained in the presence or absence of active ion channels. In real neurons, all these measurements are affected by active ion channels; so, ideally, these are also active properties, not passive! Also, please mention that in the presence of resonating channels (e.g., HCN, M-type K+), a single exponential fit won't be appropriate to obtain tau, given the presence of sag.

We thank the reviewer for pointing out this ambiguity. What the term “Passive” means in Table 1 (e.g., for the input resistance, R_in) is that the minimal set of parameters needed to validate R_in are the passive ones (i.e., Cm, Ra, and Leak). We have changed the table listing to reflect this.

**Reviewer #2 (Recommendations for the authors):**
(1) Figure 2B and the caption to Figure 2F show and describe the diameter of the sections, whereas the image in Figure 2F shows the radius. Which is the correct one?

The reason for this is that Figure 2B shows the sections' geometry as it is represented in NEURON, i.e., with diameters, while Figure 2F shows the geometry as it is represented in an SWC file (as these changes are made based on the SWC file). Nevertheless, as mentioned earlier, we decided to remove panel F from the figure in the new version, to present a more important panel on tree graph representations.

(2) "Each segment can be viewed as an equivalent RC circuit representing a part of the membrane". The example in Figure 2B is perhaps a relatively simple case. For more complex cases where multiple nonlinear conductances are present in each section, would it be possible to show each of these conductances explicitly? If yes, it would be nice to illustrate that.

We would like to clarify that "can be viewed" here was intended to mean "can be considered," and we have updated the text accordingly. The schematic RC circuits were added to the corresponding figure for illustration purposes only and are not present in the GUI, as this would indeed be impractical for multiple conductances.

(3) Some extra citations could be added. For example, it is a little strange that BRIAN2 is mentioned, but NEST is not. It might be worth mentioning and citing it. Also, the Allen Cell Types Database is mentioned, but no citation for it is given. It could be useful to add such citations (https://doi.org/10.1038/s41593-019-0417-0, https://doi.org/10.1038/s41467-017-02718-3).

Brian 2 is extensively used in our lab on its own and as a foundation of the Dendrify library (Pagkalos et al., 2023). As stated in the discussion, we are considering bridging reduced Hodgkin-Huxley-type models to Dendrify leaky integrate-and-fire type models. For these reasons, Brian 2 is mentioned in the discussion. However, we acknowledge that our previous overview omitted references to some key software, which have now been added to the updated manuscript. We appreciate the reviewer providing references that we had overlooked.

(3) Pagkalos, M., Chavlis, S. & Poirazi, P. Introducing the Dendrify framework for incorporating dendrites to spiking neural networks. Nat Commun 14, 131 (2023). https://doi.org/10.1038/s41467-022-35747-8